# High and stable ATP levels prevent aberrant intracellular protein aggregation in yeast

Masak Takaine[1,2]*, Hiromi Imamura[3], Satoshi Yoshida[1,2,4,5]*

[1]Gunma University Initiative for Advanced Research (GIAR), Gunma University, Maebashi, Japan; [2]Institute for Molecular and Cellular Regulation (IMCR), Gunma University, Maebashi, Japan; [3]Graduate School of Biostudies, Kyoto University, Kyoto, Japan; [4]School of International Liberal Studies, Waseda University, Tokyo, Japan; [5]Japan Science and Technology Agency, PREST, Tokyo, Japan

**Abstract** Adenosine triphosphate (ATP) at millimolar levels has recently been implicated in the solubilization of cellular proteins. However, the significance of this high ATP level under physiological conditions and the mechanisms that maintain ATP remain unclear. We herein demonstrated that AMP-activated protein kinase (AMPK) and adenylate kinase (ADK) cooperated to maintain cellular ATP levels regardless of glucose levels. Single-cell imaging of ATP-reduced yeast mutants revealed that ATP levels in these mutants underwent stochastic and transient depletion, which promoted the cytotoxic aggregation of endogenous proteins and pathogenic proteins, such as huntingtin and α-synuclein. Moreover, pharmacological elevations in ATP levels in an ATP-reduced mutant prevented the accumulation of α-synuclein aggregates and its cytotoxicity. The present study demonstrates that cellular ATP homeostasis ensures proteostasis and revealed that suppressing the high volatility of cellular ATP levels prevented cytotoxic protein aggregation, implying that AMPK and ADK are important factors that prevent proteinopathies, such as neurodegenerative diseases.

**\*For correspondence:**
masaktakaine@gunma-u.ac.jp (MT);
satosh@waseda.jp (SY)

**Competing interest:** The authors declare that no competing interests exist.

## Editor's evaluation

Here the authors apply a live cell imaging technique to monitor changes in $[ATP]_{cyto}$ at the single cell level in yeast. This led to the discovery of marked stability of $[ATP]_{cyto}$ (at millimolar concentration) in wildtype yeast and wild fluctuations in $[ATP]_{cyto}$ in yeasts mutant in AMP kinase and adenylate kinase. The latter fluctuations included deep dips in $[ATP]_{cyto}$, which correlated with enhanced accumulation of model abnormal human disease-associated proteins (α-Synuclein, huntingtin etc). The paper is remarkable for suggesting an important link between cellular energetics and protein folding homeostasis, which may be broadly applicable to cells of diverse phyla.

## Introduction

Adenosine triphosphate (ATP) is the main energy currency used by all living organisms. In addition to its role as energy currency, ATP has recently been suggested to influence the balance between the soluble and aggregated states of proteins, indicating that proteostasis is maintained by energy-dependent chaperones and also by the property of ATP as a hydrotrope to solubilize proteins (*Hayes et al., 2018*; *Patel et al., 2017*; *Pu et al., 2019*; *Sridharan et al., 2019*). Furthermore, ATP levels have been shown to regulate the physicochemical properties of the cytoplasm, such as viscosity, macromolecular crowding, and liquid–liquid phase separation (*Marini et al., 2020*; *Persson et al., 2020*). However, the role of ATP was assessed in these studies using artificial ATP depletion. Therefore, it

**eLife digest** Cells use a chemical called adenosine triphosphate (ATP) as a controllable source of energy. Like a battery, each ATP molecule contains a specific amount of energy that can be released when needed. Cells just need enough ATP to survive, but most cells store a lot more than they need. It is unclear why cells keep so much ATP, or whether this excess ATP has any other purpose.

To answer these questions, Takaine et al. identified mutants of the yeast *Saccharomyces cerevisiae* that had low levels of ATP, and studied how these cells differ from normal yeast The results showed that, in *S. cerevisiae* cells with lower and variable levels of ATP, proteins stick together, forming clumps. Proteins are molecules that perform diverse roles, keeping cells alive. When they clump together, they stop working and can cause cells to die. Further experiments showed that reducing the levels of ATP just for a short time increased the rate at which proteins stick together.

Taken together, Takaine et al.'s results suggest that ATP plays a role in stopping proteins from sticking together, explaining why cells may store excess ATP, since it could aid survival.

Protein clumps, also called aggregates, are a key feature of various illnesses, including neuro-degenerative diseases such as Alzheimer's. Takaine et al. provide a possible cause for why proteins aggregate in these diseases, which may be worth further study.

currently remains unclear whether ATP-dependent protein solubilization/desolubilization have physiologically significant cellular roles.

We recently established a reliable imaging technique to quantify intracellular ATP levels in single living yeast cells using the genetically encoded fluorescent ATP biosensor QUEEN (*Yaginuma et al., 2014*), which enables long-term evaluations of ATP homeostasis in individual cells (*Takaine et al., 2019*). The findings obtained demonstrated that intracellular ATP levels did not vary within a yeast population grown in the same culture (*Takaine et al., 2019*), which was in contrast to the large variations observed in intracellular ATP levels within a bacterial cell population (*Yaginuma et al., 2014*). Moreover, intracellular ATP levels in individual living yeast cells were stably and robustly maintained at approximately 4 mM, irrespective of carbon sources and cell-cycle stages, and temporal fluctuations in intracellular ATP levels were small (*Takaine et al., 2019*). Based on these findings, we hypothesized that an exceptionally robust mechanism exists to precisely regulate ATP levels in eukaryotes. It currently remains unclear why ATP is stably maintained at a markedly higher level than the $K_m$ (Michaelis constant) required for the enzymatic activities of most ATPases (*Edelman et al., 1987*), and the consequences associated with failed ATP homeostasis in living organisms have not yet been elucidated.

The most promising candidate regulator of ATP homeostasis is adenosine monophosphate (AMP)-activated protein kinase (AMPK). AMPK, which is activated by AMP and inhibited by ATP (*Xiao et al., 2007*), has long been regarded as an important regulator of the whole-body and cellular energy status in eukaryotes (*Hardie et al., 2016*). AMPK is activated by increases in the AMP:ATP or ADP:ATP ratio (i.e., low-energy state), and regulates its downstream effectors by phosphorylation to redirect cell metabolism from an anabolic (ATP-consuming) state to catabolic (ATP-producing) state (*Herzig and Shaw, 2018*). In the budding yeast *Saccharomyces cerevisiae*, the sucrose nonfermenting 1 (Snf1) protein kinase complex is the sole AMPK. Similar to other AMPKs, the yeast Snf1 complex comprises three subunits: the catalytic α subunit (*SNF1*), scaffolding β subunit (*SIP1*, *SIP2*, or *GAL83*), and regulatory γ subunit (*SNF4*) (*Ghillebert et al., 2011*). The role of the Snf1 complex in adaptation to glucose limitations has been characterized in detail (*Hedbacker and Carlson, 2008*). The Snf1 complex is inactive in the presence of sufficient glucose levels in media (*Wilson et al., 1996*). Decreases in glucose levels have been shown to activate the Snf1 complex and phosphorylate the transcriptional repressor Mig1, which then triggers the transcription of numerous glucose-repressed genes (*Carlson, 1999*). However, the contribution of AMPK or the Snf1 complex to cellular ATP levels remains unknown.

Other possible candidate regulators of ATP homeostasis include genes whose mutation leads to decrease in the cellular content of ATP. However, based on biochemical analyses of cell populations, few yeast mutants reduced ATP levels (*Gauthier et al., 2008*; *Ljungdahl and Daignan-Fornier, 2012*). Adenylate kinase (ADK) is a key enzyme that synthesizes ATP and AMP using two adenosine diphosphate (ADP) molecules as substrates, and the null mutant of ADK (*adk1Δ*) was shown to have a reduced

cellular ATP level (~70% of the wild type) (*Gauthier et al., 2008*). Bas1 is a transcription factor that is required for de novo purine synthesis and *bas1Δ* also has a reduced ATP level (~50% of the wild type) (*Gauthier et al., 2008*). However, the regulation of ATP levels and the physiological consequences of reduced ATP levels in these mutants remain unclear, particularly at the single-cell level.

In the present study, we investigated the roles of AMPK, ADK, and Bas1 in ATP homeostasis using the QUEEN-based single-cell ATP imaging technique. We demonstrated for the first time that AMPK is involved in the regulation of intracellular ATP levels, even under glucose-rich conditions. Furthermore, time-lapse ATP imaging revealed that cells lacking both AMPK and ADK frequently underwent transient ATP depletion, while ATP levels oscillated in those lacking Bas1. These ATP dynamics in the mutants were overlooked in previous biochemical studies. The transient depletion of ATP closely correlated with and often preceded the accelerated accumulation of protein aggregates. We found that some intrinsic proteins and aggregation-prone model proteins, including α-synuclein, which is responsible for Parkinson's disease, aggregated and were cytotoxic in all of the ATP-reduced mutants tested. The present results suggest that the stable maintenance of ATP is essential for proteostasis and imply that an ATP crisis promotes proteinopathies, such as neurodegenerative diseases.

## Results

### ADK1 cooperates with AMPK to regulate ATP homeostasis

We recently developed a reliable monitoring system for cytoplasmic ATP levels in living yeast cells using the ATP biosensor QUEEN (*Takaine et al., 2019*). We herein conducted a more detailed examination of ATP dynamics in wild-type and mutant yeast cells using this system. We initially investigated whether the deletion of *SNF1*, which encodes a catalytic subunit of AMPK, affected cellular ATP levels. Cellular ATP levels were significantly lower in *SNF1*-null mutant (*snf1Δ*) cells than in wild-type cells at various glucose levels (*Figure 1A* and *Figure 1—figure supplement 1A, B*). It is important to note that in addition to low glucose conditions (0.05% glucose), at which the Snf1 complex is active, *snf1Δ* cells showed significantly reduced ATP levels even under high glucose conditions (2% glucose), at which the Snf1 complex is considered to be inactive. Moreover, the deletion of each gene encoding the β subunit of AMPK (*SIP1*, *SIP2*, or *GAL83*) reduced cellular ATP levels to a similar extent as the deletion of *SNF1*, suggesting that three subtypes of the Snf1 complex were involved in maintaining ATP concentrations (*Figure 1—figure supplement 1C*). On the other hand, the deletion of *MIG1* had a negligible effect on ATP levels (*Figure 1—figure supplement 1D*), suggesting that as yet unknown factors other than Mig1 primarily regulate ATP levels under the control of the Snf1 complex. Collectively, these results demonstrated for the first time that AMPK/SNF1 affect cellular ATP levels even under glucose-rich conditions.

ADK catalyzes the interconversion of adenine nucleotides (ATP + AMP ⟷ 2ADP), which is important for de novo adenine nucleotide synthesis and the balance between ATP, ADP, and AMP. Previous biochemical studies reported that the deletion of the ADK gene-reduced ATP levels in mouse skeletal muscle cells and budding yeasts (*Gauthier et al., 2008*; *Janssen et al., 2000*). We confirmed these findings using an ATP imaging method: *adk1Δ* cells showed significantly lower QUEEN ratios than wild-type cells on average in the presence of sufficient carbon sources (*Figure 1—figure supplement 2*).

In addition to being a key enzyme in purine metabolism, ADK has also been suggested to cooperate with AMPK in order to monitor the cellular energy state (*Hardie et al., 1998*). Therefore, we compared ATP levels in *snf1Δ adk1Δ* double mutant cells with those in *snf1Δ*, *adk1Δ* cells, and wild-type cells (*Figure 1A, B*). *snf1Δ adk1Δ* cells had significantly lower average ATP levels than single mutant cells. We also found not only a general reduction, but also a marked variation in ATP levels in the *snf1Δ*, *adk1Δ*, *snf1Δ adk1Δ* cell population, as indicated by the large coefficient of variance (CV) (*Figure 1B*). Furthermore, some *snf1Δ adk1Δ* cells had very low ATP levels (*Figure 1A, B*). These results suggest that Adk1 and the Snf1 complex both synergistically contribute to ATP homeostasis.

We confirmed the decreases observed in ATP levels in *snf1Δ*, *adk1Δ*, and *snf1Δ adk1Δ* cells using a biochemical assay of whole cell extract (*Figure 1C*). ADP levels were also reduced in these mutant cells (*Figure 1—figure supplement 3A*). In *adk1Δ* cells, the ATP/ADP ratio increased, whereas the sum of ATP and ADP levels decreased (*Figure 1—figure supplement 3C,D*), which is consistent with

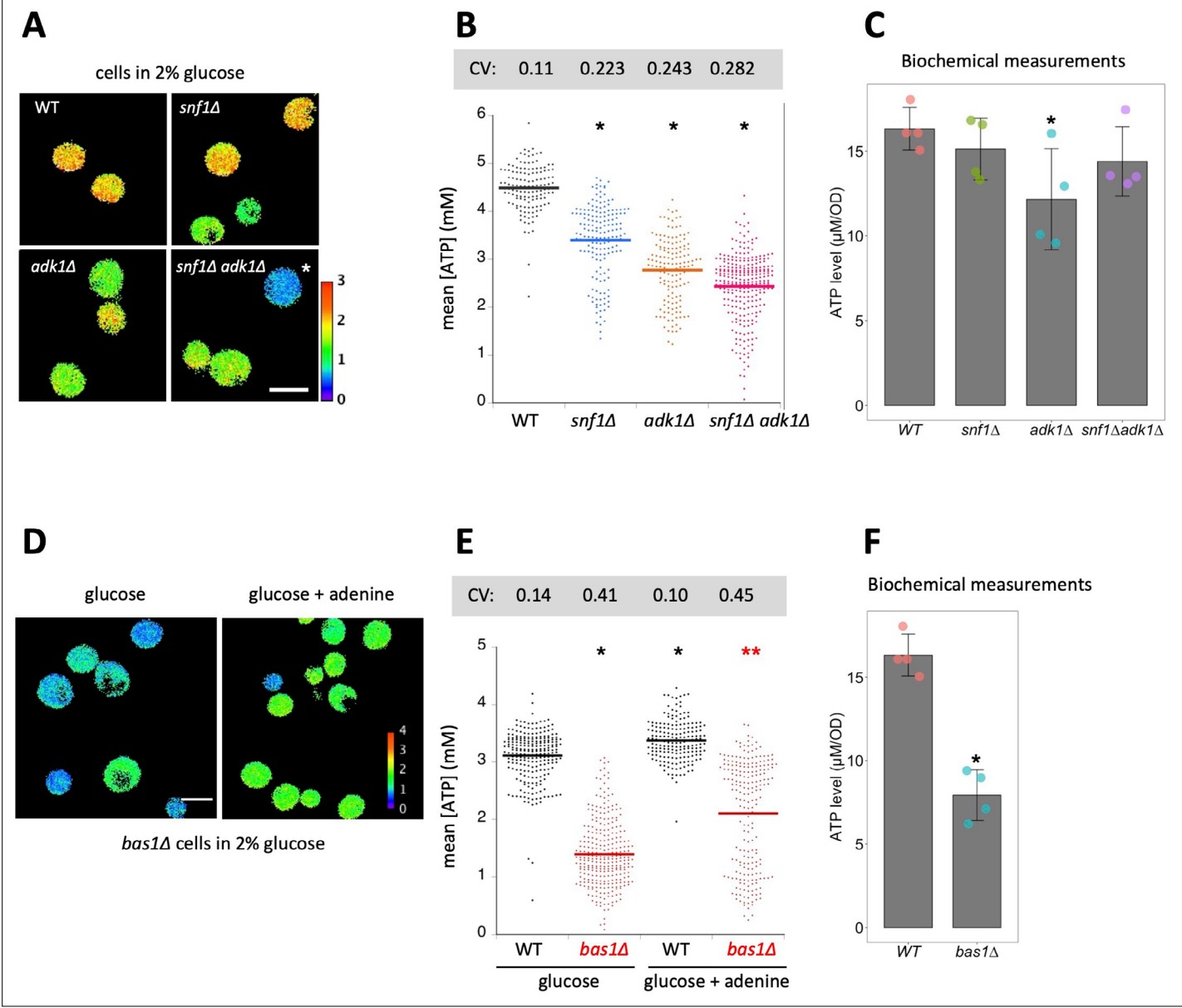

**Figure 1.** Interconversion and active synthesis of adenine nucleotides are important for adenosine triphosphate (ATP) homeostasis. (**A**) Adk1 and Snf1 synergistically control cellular ATP levels. QUEEN ratio images of ATP homeostasis mutant cells grown in medium containing 2% glucose. The asterisk indicates an example of cells with significantly reduced ATP levels. (**B**) The mean QUEEN ratios of cells were translated to ATP levels and shown in a dot plot. The horizontal bar indicates the mean of each population. Asterisks indicate p values less than 0.0001 versus WT (Dunnett's multiple comparison). CV: coefficient of variance. $N$ = 134–276 cells were scored. (**C**) Biochemical measurements of cellular ATP levels. ATP levels in cells of the indicated genotypes were measured as described in Materials and methods. Data are the mean ± 1SD (error bars) ($N$ = 4). An asterisk indicates a p value of 0.022 versus WT (Dunnett's multiple comparison). (**D**) QUEEN ratio images of $bas1\Delta$ cells grown in 2% glucose medium. Growth in media supplemented with 0.11 mg/ml adenine partially restored the low ATP phenotype of $bas1\Delta$. (**E**) ATP levels in cells shown in D were plotted. Asterisks indicate p values: *p < 0.0001 (versus WT in glucose, Dunnett's multiple comparison); **p = $3.6 \times 10^{-20}$ (versus $bas1\Delta$ in glucose). $N$ = 186–296 cells were scored. (**F**) ATP levels in WT and $bas1\Delta$ cells were measured as described in C. Data are the mean ± 1SD (error bars) ($N$ = 4). An asterisk indicates a p value of $8.5 \times 10^{-5}$ versus WT.

The online version of this article includes the following source data and figure supplement(s) for figure 1:

**Source data 1.** Raw data for *Figure 1*.

**Figure supplement 1.** AMP-activated protein kinase (AMPK) is involved in the maintenance of cellular adenosine triphosphate (ATP) levels in nonstarving cells.

**Figure supplement 1—source data 1.** Raw data for *Figure 1—figure supplement 1*.

*Figure 1 continued on next page*

*Figure 1 continued*

**Figure supplement 2.** Adenylate kinase Adk1 is involved in the maintenance of cellular adenosine triphosphate (ATP) levels.

**Figure supplement 2—source data 1.** Raw data for *Figure 1—figure supplement 2*.

**Figure supplement 3.** Biochemical measurements of adenosine triphosphate (ATP) and ADP.

**Figure supplement 3—source data 1.** Raw data for *Figure 1—figure supplement 3*.

**Figure supplement 4.** Adenosine triphosphate (ATP) levels in *bas1Δ snf1Δ* cells.

**Figure supplement 4—source data 1.** Raw data for *Figure 1—figure supplement 4*.

previous findings (*Gauthier et al., 2008*). These biochemical data and their relevance to QUEEN data are discussed later.

## A large pool of adenine nucleotides is important for maintaining cellular ATP levels

We examined a *bas1Δ* mutant, which is defective in the expression of genes responsible for adenine biogenesis (*Daignan-Fornier and Fink, 1992*; *Denis et al., 1998*). Consistent with previous biochemical findings (*Gauthier et al., 2008*), ATP levels quantified by QUEEN were reduced by ~50% in *bas1Δ* cells (*Figure 1D, E*). We found not only a general reduction, but also a marked variation in ATP levels in the *bas1Δ* cell population, as indicated by the large CV (*Figure 1E*). The decrease observed in ATP levels was due to reduced adenine biosynthesis because the addition of extra adenine to media partially restored ATP levels (*Figure 1E*). These results suggest that the sufficient production of adenine nucleotides is essential for the stable maintenance of ATP levels. Moreover, the role of Bas1 in maintaining ATP levels appeared to be epistatic to that of Snf1 because *bas1Δ snf1Δ* double mutant cells showed a similar distribution of ATP levels to *bas1Δ* cells (*Figure 1—figure supplement 4*). We confirmed the decrease observed in ATP levels in *bas1Δ* cells using a biochemical assay (*Figure 1F*). ADP levels and the sum of ATP and ADP levels also significantly decreased in *bas1Δ* cells (*Figure 1— figure supplement 3A,D*), which is consistent with previous findings (*Gauthier et al., 2008*).

## ATP levels temporally fluctuate in ATP-mutant cells

To investigate the mechanisms contributing to the marked variations in ATP levels in *snf1Δ adk1Δ* cells in more detail, we employed time-lapse ATP imaging (*Figure 2*). In wild-type cells, the QUEEN ratio fluctuated in a narrow range as previously reported (*Figure 2A*; *Takaine et al., 2019*). We found that the QUEEN ratio underwent a rapid decline followed by recovery in *snf1Δ adk1Δ* cells (see 116 and 132 min in *Figure 2B, D*, and *Figure 2—video 1*, and 180 and 356 min in *Figure 2C, E*, and *Figure 2—video 2*). The sudden decrease in ATP levels (hereafter called 'the ATP dip', see Materials and methods for the detailed definition) occurred within a few minutes without any sign and was scarcely observed in wild-type cells (*Figure 2A*). Under some conditions, the QUEEN ratio remained at a lower level after the decrease, and did not recover within the observation time (*Figure 2—figure supplement 1*). We defined this case as 'the ATP shift'. In total, 35% of *snf1Δ adk1Δ* cells exhibited the ATP shift or ATP dip, whereas 99% of wild-type cells showed stable ATP dynamics (*Figure 2F*). The ATP dip and shift appeared to be a stochastic event that is intrinsic to cells; these events occurred independent of the cell-cycle stage or cell size (compare *Figure 2D, E*). Moreover, a correlation was not observed between the duration of the ATP dip and the extent of the decrease in the QUEEN ratio (*Figure 2—figure supplement 2C*). These results suggest that the marked variations observed in ATP levels in *snf1Δ adk1Δ* cells were not simply due to a mixed population with different basal ATP levels, but were rather caused by the stochastic ATP dip in individual cells.

Time-lapse imaging of *bas1Δ* revealed oscillatory cycles in ATP levels (*Figure 2G* and *Figure 2— figure supplement 3A*, and *Figure 2—video 3* and *Figure 2—video 4*): ATP cycling in *bas1Δ* cells was slow (~35 min on average, *Figure 2—figure supplement 3B*) and distinguishable from that in *snf1Δ adk1Δ*cells; however, the common characteristics of these mutants were that the level of ATP often reached close to 0 mM. The ATP oscillation cycle was unsynchronized in the population and independent of cell-cycle progression, suggesting a unique metabolic rhythm intrinsic to each cell. The oscillatory nature of ATP cycling in the *bas1Δ* mutant may involve a transcription/translation cycle and will be described elsewhere.

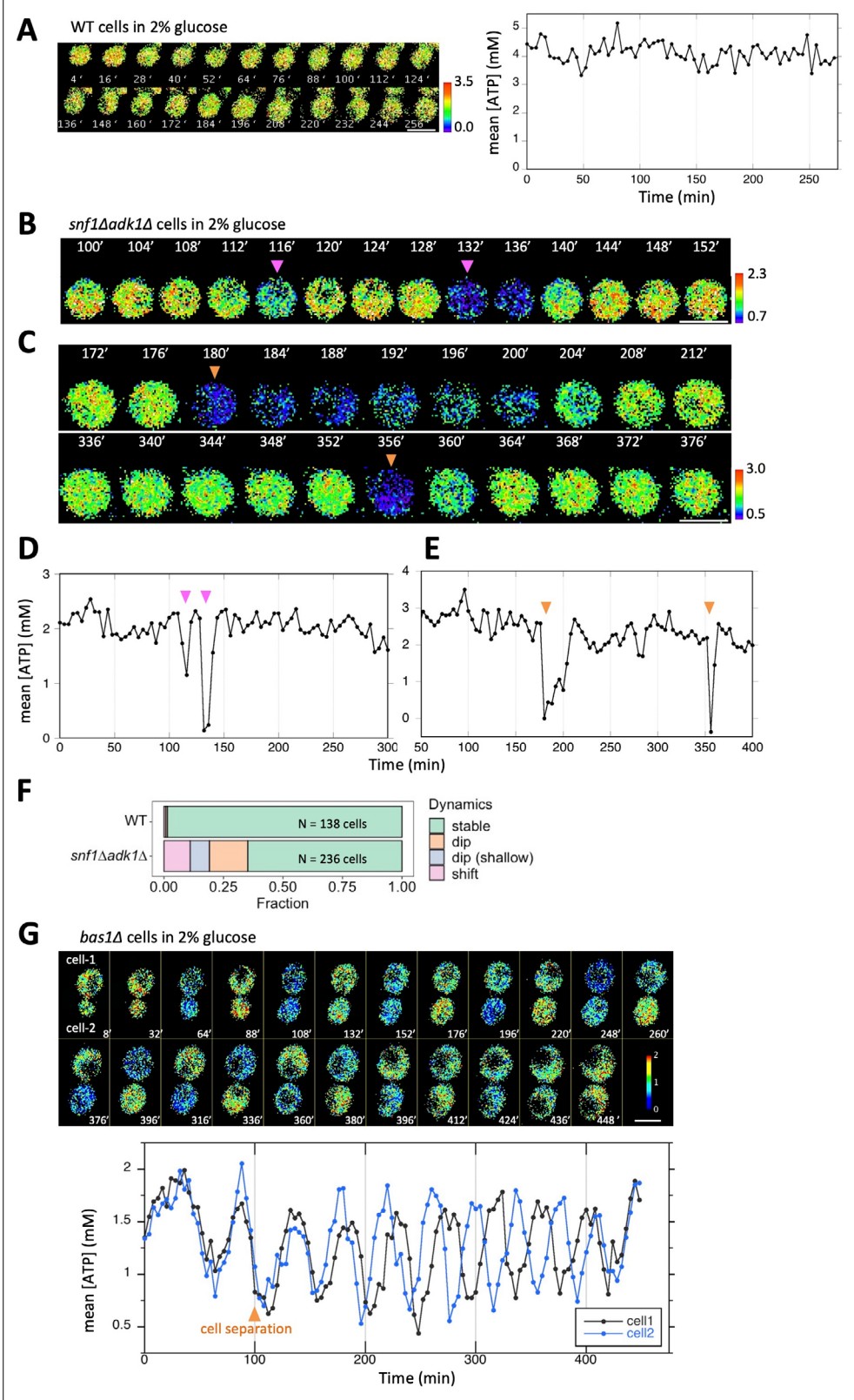

**Figure 2.** Temporal fluctuations in adenosine triphosphate (ATP) levels in *snf1Δ adk1Δ* and *bas1Δ* cells. (**A**) Time-lapse imaging of QUEEN in wild-type cells cultured in 2% glucose medium. QUEEN ratios were converted into intracellular ATP levels in cells and plotted (right panel). (**B, D**) Time-lapse imaging of QUEEN in *snf1Δ adk1Δ* cells. Images at the representative time points were shown. The QUEEN ratio decreased twice (indicated by arrowheads)

*Figure 2 continued on next page*

*Figure 2 continued*

within a short interval. See also *Figure 2—video 1*. Data were converted into ATP levels and plotted in (D). (C, E) Another example of the time-lapse imaging of QUEEN in *snf1Δ adk1Δ* cells in 2% glucose medium. The QUEEN ratio decreased twice (indicated by arrowheads) with a long interval (a rare event). See also *Figure 2—video 2*. Data were converted into ATP levels and plotted in (E). (F) Some *snf1Δ adk1Δ* cells showed unstable ATP levels. Wild-type and *snf1Δ adk1Δ* cells were classified and scored according to the indicated ATP dynamics. Data were pooled from three independent experiments for each strain. (G) Time-lapse imaging of QUEEN in *bas1Δ* in 2% glucose medium. ATP levels in the mother (cell-1) and daughter (cell-2) were plotted at the bottom. Images at the representative time points were shown on the top. Note that the QUEEN ratio is synchronized until cells undergo separation at the time point of 76 min indicated by an arrow. After separation, each cell has a unique periodic cycle of ATP. The movie is available in *Figure 2—video 3*. White scale bar = 5 μm.

The online version of this article includes the following video, source data, and figure supplement(s) for figure 2:

**Source data 1.** Raw data for *Figure 2*.

**Figure supplement 1.** Time-lapse imaging of QUEEN in an *snf1Δ adk1Δ* cell showing the adenosine triphosphate (ATP) shift.

**Figure supplement 1—source data 1.** Raw data for *Figure 2—figure supplement 1*.

**Figure supplement 2.** Temporal fluctuations in the QUEEN ratio in wild-type and *snf1Δ adk1Δ* cells.

**Figure supplement 2—source data 1.** Raw data for *Figure 2—figure supplement 2*.

**Figure supplement 3.** Oscillatory behavior of the adenosine triphosphate (ATP) level visualized in *bas1Δ* cells.

**Figure supplement 3—source data 1.** Raw data for *Figure 2—figure supplement 3*.

**Figure 2—video 1.** Time-lapse imaging of QUEEN in *snf1Δ adk1Δ* cells in 2% glucose medium. https://elifesciences.org/articles/67659/figures#fig2video1

**Figure 2—video 2.** Another example of *snf1Δ adk1Δ* cells showing a sudden decrease in the QUEEN ratio. https://elifesciences.org/articles/67659/figures#fig2video2

**Figure 2—video 3.** Oscillatory behavior of the QUEEN ratio in *bas1Δ* cells. https://elifesciences.org/articles/67659/figures#fig2video3

**Figure 2—video 4.** Another example of *bas1Δ* cells showing an oscillating QUEEN ratio. https://elifesciences.org/articles/67659/figures#fig2video4

## ATP homeostasis is required for preventing protein aggregation in vivo

We recently reported that cellular ATP levels were stably maintained at ~4 mM in budding yeast (*Takaine et al., 2019*) and herein demonstrated that Adk1 and Bas1 in the Snf1/AMPK complex were required for the regulation of ATP homeostasis. However, the physiological importance of ATP homeostasis remains unknown. To clarify the significance of high ATP levels, we examined the global genetic interactions of *snf1Δ, adk1Δ*, and *bas1Δ* using CellMap (*Usaj et al., 2017*, thecellmap.org). An in silico analysis identified genes involved in 'protein folding/glycosylation' as common negative genetic interactors with *adk1Δ* and *bas1Δ* (*Figure 3A*). Negative genetic interactors of *ura6*, a gene encoding uridylate kinase that also exhibits ADK activity, were enriched in the 'protein folding/glycosylation' category (*Figure 3A*). We also found that interactors of *snf1* were implicated in 'protein folding/glycosylation'. None of these mutants exhibited apparent genetic interactions with genes in the 'metabolism' category (*Figure 3A*). The same analysis using genetic and physical interactors provided similar results and showed that many interactors were enriched in the 'protein turnover' category (*Figure 3—figure supplement 1A*). These results imply that although these three mutants regulate ATP with distinct mechanisms, all three have a common cellular function.

To examine possible defects in protein folding and turnover (i.e., proteostasis), we challenged these mutants with various proteotoxic stresses. We found negligible growth defects in ATP mutants under normal growth conditions with 2% glucose at 30°C (control in *Figure 3B*), suggesting that a high level of ATP is not necessary for cellular growth. However, the *adk1* and *bas1* mutants both exhibited severe growth defects with a high temperature of 40°C, 1 hr of heat shock at 55°C, or in the presence of 0.5 μg/ml of the glycosylation inhibitor tunicamycin or 2 mM $H_2O_2$, an inducer of oxidative stress. The *SNF1* deletion increased the stress sensitivity of *adk1Δ* (*Figure 3B*). This sensitivity to proteotoxic stress suggests that ATP homeostasis mutants are defective in some aspects of proteostasis. We found that all four mutants tested contained significantly increased numbers of

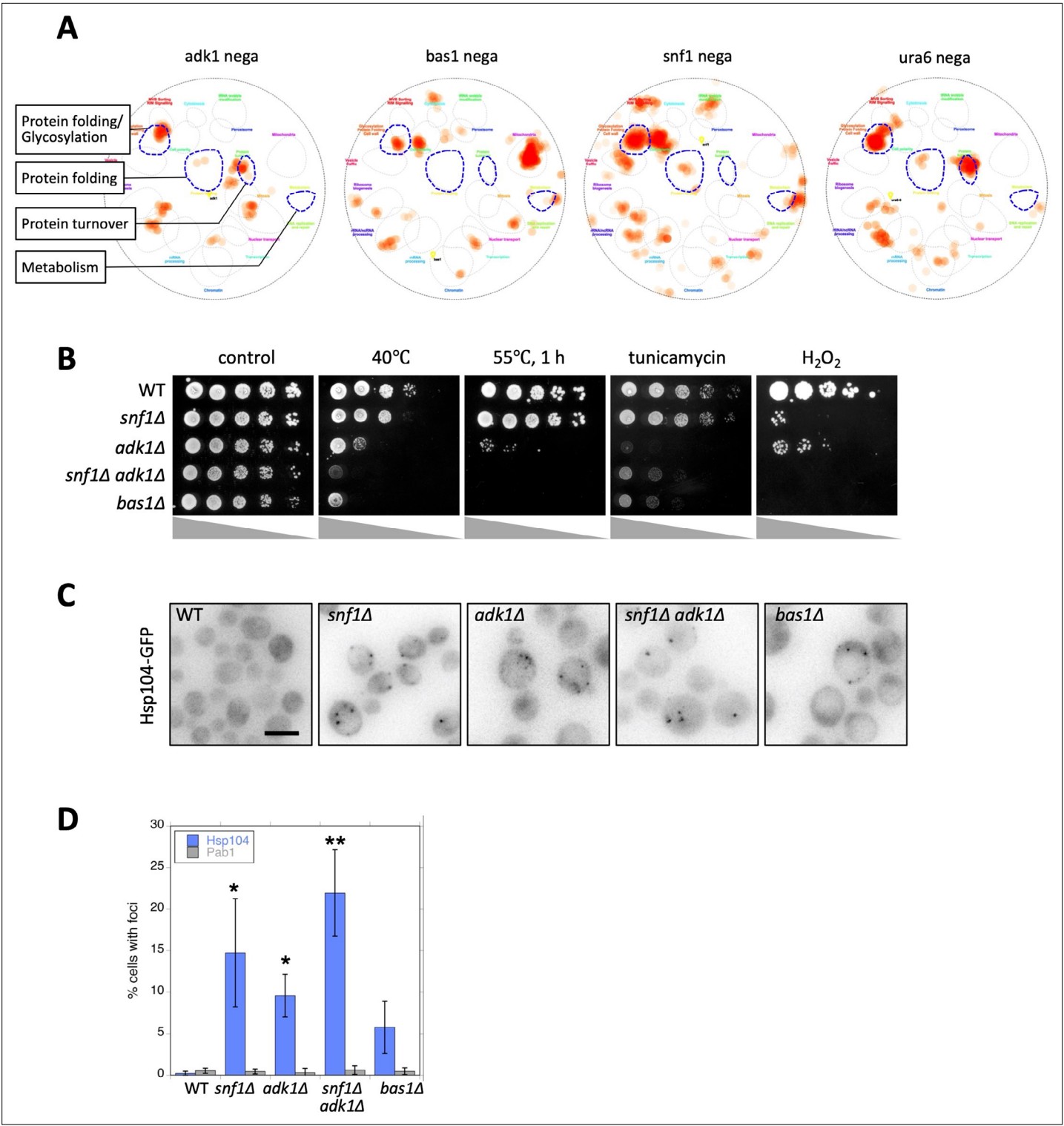

**Figure 3.** Adenosine triphosphate (ATP) homeostasis is required to prevent protein aggregation. (**A**) Functional landscape of known interactors of ATP mutants. Negative genetic interactors of the indicated gene were derived from the SGD database (https://www.yeastgenome.org/; *Cherry et al., 2012*) and overlaid on a functional map based on the global genetic interaction network of the yeast genome (*Baryshnikova, 2016*; *Usaj et al., 2017*). *URA6* encodes an uridylate kinase that is essential for viability, which also exhibits adenylate kinase activity. Regarding information on all the categories of functions, refer to *Figure 3—figure supplement 1B*. (**B**) Each strain of the indicated genotype was serially diluted (fivefold), spotted on SC + 2% glucose medium, and grown under the indicated stress conditions. Photos were taken after 2–3 days. (**C**) Formation of Hsp104-GFP foci in ATP homeostasis mutants. The GFP signal (inverted grayscale) was imaged in the log-phase culture of the indicated mutant cells expressing Hsp104-GFP

*Figure 3 continued on next page*

*Figure 3 continued*

at 35°C. (**D**) Quantification of data shown in (**C**). Data from similar experiments using strains expressing Pab1-GFP, instead of Hsp104-GFP, were also plotted. Values are the mean ± 1SD (error bars) (*N* = 4). Asterisks indicate a significant difference from WT: *$p < 0.05$; **$p < 0.001$, Dunnett's multiple comparison. White scale bar = 5 µm.

The online version of this article includes the following source data and figure supplement(s) for figure 3:

**Source data 1.** Raw data for *Figure 3*.

**Figure supplement 1.** In silico analysis of interactors of adenosine triphosphate (ATP) mutants.

Hsp104-GFP foci, a marker of protein aggregation (*Kirkwood et al., 2017*; *Figure 3C, D*). In contrast to Hsp104-GFP foci, Pab1-GFP, a marker of stress granule (SG) assembly (*Hoyle et al., 2007*), did not form foci in ATP mutants, suggesting that protein aggregation and SG assembly are regulated in a distinct manner (*Figure 3D*). These analyses identified abnormal protein aggregation as a common defect associated with ATP homeostasis mutants for the first time.

## The transient depletion of ATP leads to the formation of protein aggregates

To examine whether ATP depletion triggers protein aggregation in living yeast, we artificially depleted cellular ATP levels by replacing glucose with 2-deoxyglucose (2DG), a strong inhibitor of glycolysis, in media and monitored protein aggregation using Hsp104-GFP as a marker of protein aggregation (*Kirkwood et al., 2017*) in wild-type cells. We previously showed that ATP levels were almost completely depleted 2 min after the 2DG treatment (*Takaine et al., 2019*), which was also confirmed biochemically (*Figure 1—figure supplement 3B*). Within 15 min of the 2DG treatment, more than 20% of cells contained Hsp104-GFP foci indicative of protein aggregation (*Figure 4A, B*). These protein aggregations were retained for hours after refeeding of glucose (*Figure 4A, B*), suggesting that the dissolution kinetics of Hsp104-GFP were significantly slow. This contrasts intracellular ATP, which recovers to normal levels within 1 min of glucose refeeding (*Takaine et al., 2019*).

To further confirm whether a high level of ATP is required for protein solubility, we also tested the Ubc9-ts protein, a model protein that is prone to aggregation (*Kaganovich et al., 2008*), and found that ATP depletion by the 2DG treatment triggered Ubc9-ts protein aggregation (*Figure 4C*). Therefore, not only Hsp104-GFP-positive intrinsic proteins, but also extrinsic model proteins aggregate after ATP depletion.

SG are assembled in budding yeast cells under stress conditions, such as glucose depletion (*Hoyle et al., 2007*). In contrast to the formation of Hsp104-GFP foci, ATP depletion after the 2DG treatment did not instantly trigger the formation of SG (*Figure 4—figure supplement 1*). Consistent with recent findings (*Jain et al., 2016*), the present results suggest that SG formation requires ATP. We also noted that Hsp104-GFP foci and SG did not colocalize, indicating that these structures are derived from distinct mechanisms (*Figure 4—figure supplement 1*). Therefore, the artificial depletion of ATP may trigger abnormal protein aggregation in living yeast cells.

## ATP homeostasis is required for the protection of cells from cytotoxicity caused by protein aggregation

Protein aggregation is often associated with neurodegenerative diseases, such as Alzheimer's, Huntington's, and Parkinson's diseases (*Eftekharzadeh et al., 2016*). Mitochondrial failure has also been associated with many neurodegenerative diseases; however, it currently remains unclear whether energy failure causes protein aggregation because mitochondria also produce cytotoxic reactive oxygen species (*Bhat et al., 2015*; *Pathak et al., 2013*). The abnormal aggregation of α-synuclein has been implicated in Parkinson's disease (*Lashuel et al., 2013*). To clarify whether ATP prevents the formation of cytotoxic protein aggregation, we examined the toxicity of α-synuclein-GFP (Syn-GFP) in budding yeast. As reported previously, the expression of Syn-GFP exhibited negligible toxicity against wild-type yeast when expressed under the inducible *GAL1* promotor (*Figure 5A*; *Outeiro and Lindquist, 2003*; *Sharma et al., 2006*; *Wijayanti et al., 2015*). However, *snf1Δ*, *adk1Δ*, *snf1Δ adk1Δ*, and *bas1Δ* were hypersensitive to the expression of Syn-GFP (*Figure 5A*). We also found that *rpn4Δ*, which encodes a key transcription factor for proteasomal subunits (*Xie and Varshavsky, 2001*), was very sensitive to Syn-GFP (*Figure 5A*), which is consistent with the concept that Syn-GFP is degraded

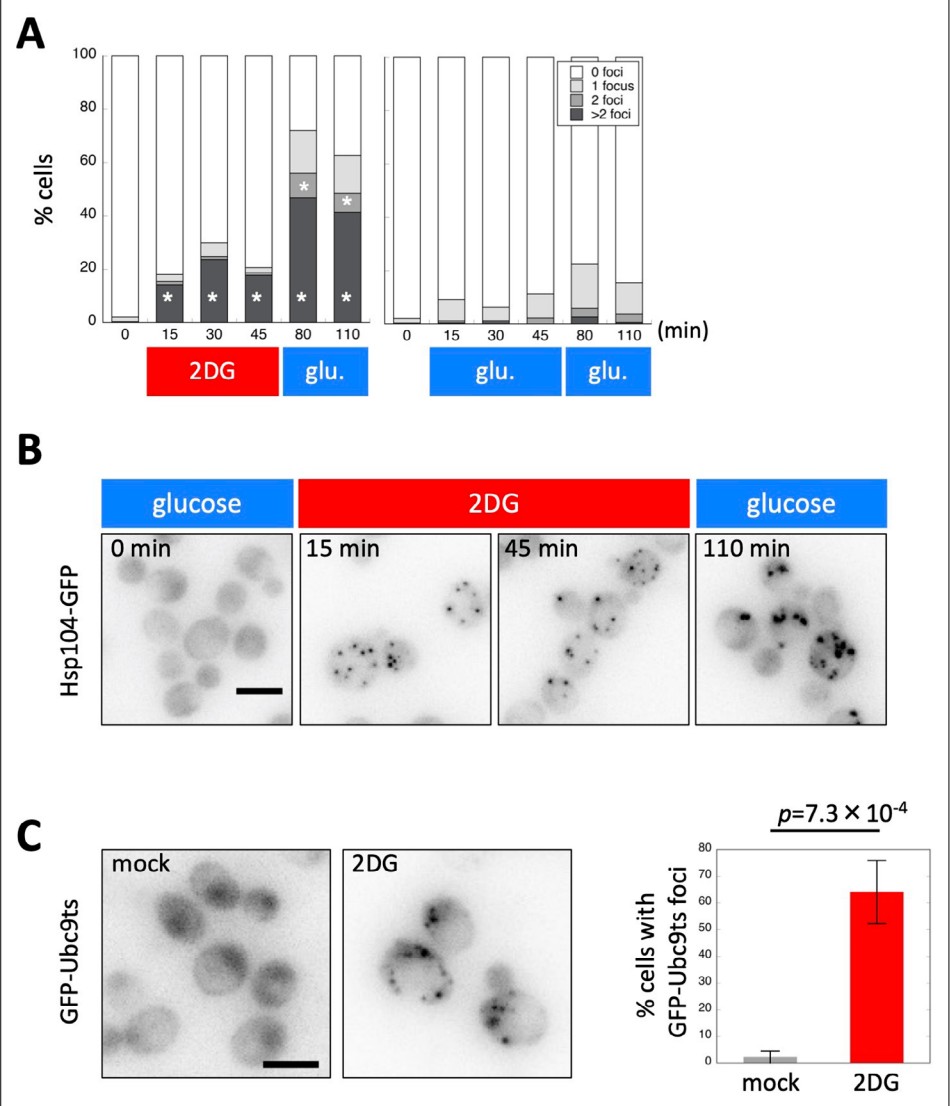

**Figure 4.** Adenosine triphosphate (ATP) depletion triggers protein aggregation in living yeast cells. (**A**) The formation of Hsp104-GFP foci after ATP depletion. Wild-type cells expressing Hsp104-GFP were grown to the log phase at 35°C in medium containing 2% glucose. At the time point of 0 min, medium was replaced with 20 mM 2-deoxyglucose (2DG; red) or 2% glucose (as a control; blue). Cells were released back to media containing 2% glucose at the time point of 50 min. Cells were imaged at the indicated time points, classified, and scored according to the number of Hsp104-GFP foci. Values are the mean (N = 3). Asterisks indicate a significant difference from the 2% glucose treatment (p < 0.05). (**B**) Representative images of cells analyzed in (**A**). (**C**) Formation of Ubc9-ts foci after ATP depletion. Cells expressing GFP-Ubc9-ts under an inducible *GAL* promoter were grown in medium containing 2% galactose (SC-gal) at 33°C, and medium was then exchanged with 2DG or SC-gal. After 30 min, cells were imaged and scored for the number of GFP-Ubc9-ts foci. Representative images (inverted grayscale) are shown on the left and summarized on the right. Values are the mean ± 1SD (error bars) (*N* = 4). White scale bar = 5 μm.

The online version of this article includes the following source data and figure supplement(s) for figure 4:

**Source data 1.** Raw data for *Figure 4*.

**Figure supplement 1.** Simultaneous observation of Hsp104 and Pab1 foci wild-type cells expressing Hsp104-GFP and Pab1-RedStar2 were grown to the log phase at 37°C in medium containing 2% glucose.

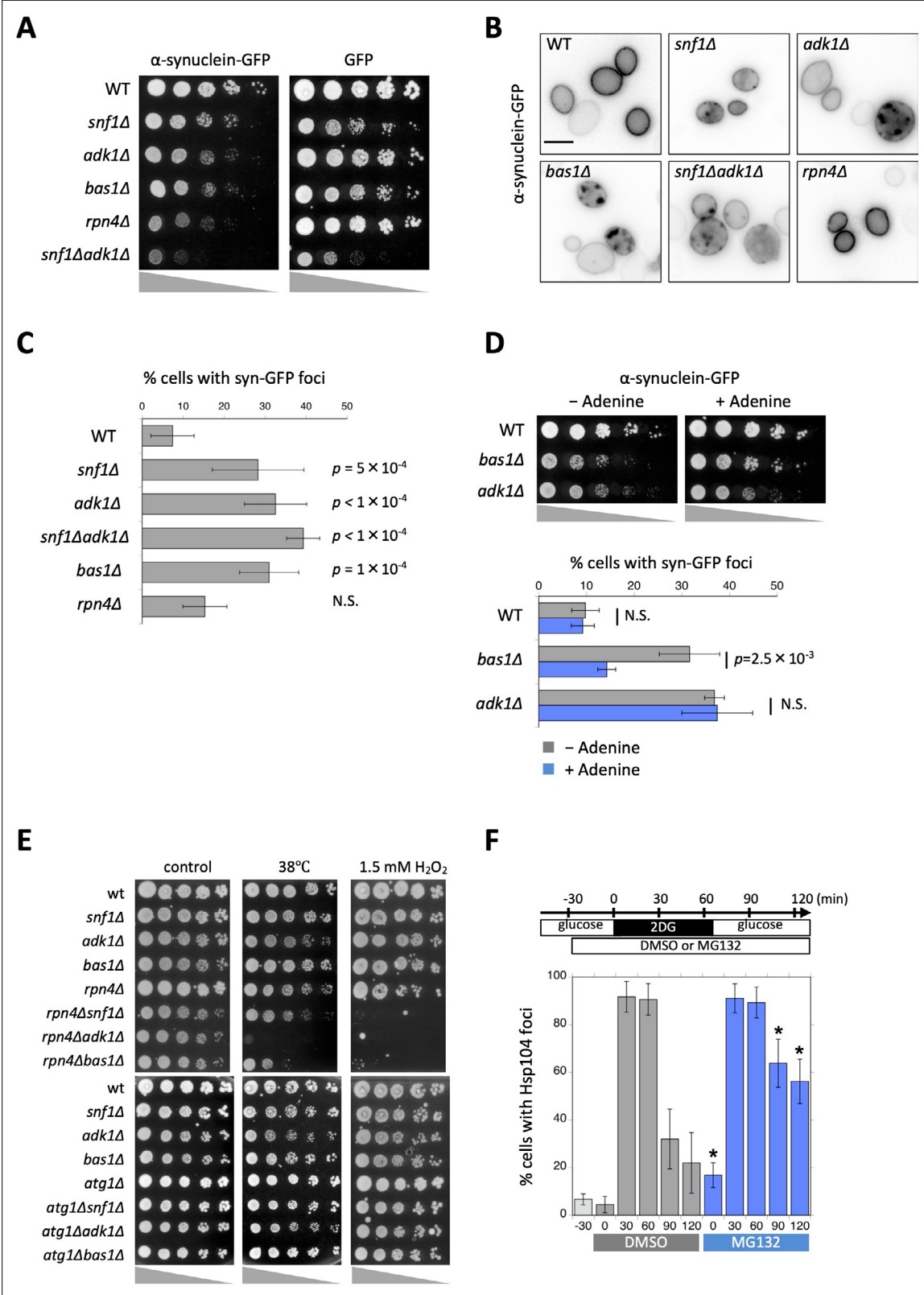

**Figure 5.** Aggregation and cytotoxicity of α-synuclein depend on adenosine triphosphate (ATP) homeostasis. (**A**) Each strain of the indicated genotype was transformed with an expression vector carrying α-synuclein-GFP or GFP, serially diluted (fivefold), spotted on SC + 2% galactose plates, and then grown at 30°C for 3 days. (**B**) The localization of α-synuclein-GFP in ATP mutants. Cells were grown on galactose plates at 30°C for more than 42 hr and then imaged. Representative images of α-synuclein-GFP (inverted grayscale) are shown. (**C**) Quantification of the data shown in (**B**). Cells were classified

*Figure 5 continued on next page*

*Figure 5 continued*

and scored for the localization pattern of α-synuclein-GFP. The percentage of cells showing α-synuclein-GFP foci are plotted. Data are the mean ± 1SD (error bars) from three to six independent observations. N = 33–380 cells were scored in each measurement. p values versus WT are shown (Dunnett's multiple comparison). N.S., no significance (p value >0.05). (**D**) (*Top*) Each strain of the indicated genotype was transformed with an expression vector carrying α-synuclein-GFP and grown on galactose plates containing 0 mM (−Adenine) or 0.3 mM (+Adenine) adenine at 30°C for 3 days. (*Bottom*) Cells were grown on galactose plates in the absence or presence of adenine at 30°C for 41–45 hr and then imaged. The percentage of cells showing α-synuclein-GFP foci was plotted. Data are the mean ± 1SD (error bars) from five independent transformants. N = 53–258 cells were scored in each measurement. Statistical significance was tested using the unpaired two-tailed Welch's *t*-test. p values versus '−Adenine' are shown. (**E**) Each strain of the indicated genotype was serially diluted (fivefold), spotted on SC + 2% glucose medium, and grown under the indicated stress conditions. (**F**) Cells of the drug-sensitive strain Y13206 were grown to the log phase at 37°C in medium containing 2% glucose and supplemented with 0.1% dimethylsulfoxide (DMSO) or 0.1% DMSO plus 42 μM MG132 at $t = -30$ min. At $t = 0$ min, these cells were washed and released in medium containing 20 mM 2-deoxyglucose (2DG) ± MG132, and cells were then washed and released again in medium containing 2% glucose ± MG132. Cells were imaged at the indicated time points and scored for the number of Hsp104-GFP foci. Data are the mean ± 1SD (error bars). Asterisks indicate a significant difference from DMSO ($p < 0.02$) ($N = 3$).

The online version of this article includes the following source data and figure supplement(s) for figure 5:

**Source data 1.** Raw data for *Figure 5*.

**Figure supplement 1.** Cytotoxicity of polyQ containing the huntingtin protein in wild-type and adenosine triphosphate (ATP)-mutant yeast cells.

in the ubiquitin–proteasomal pathway in yeast (*Tofaris et al., 2011*; *Wijayanti et al., 2015*). We then visualized the cellular localization of Syn-GFP. Consistent with previous findings (*Willingham et al., 2003*), Syn-GFP expressed in yeast mainly localized to the plasma membrane (*Figure 5B, C*). In addition to the plasma membrane, we found that Syn-GFP localized to punctate structures, most likely corresponding to protein aggregation (*Figure 5B, C*). Punctate structures were not as obvious in *rpn4Δ* cells defective in proteasomes, suggesting that the accumulation of Syn-GFP puncta was not simply due to defective degradation.

To investigate whether a high level of ATP protects cells from toxic protein aggregation, we added extra adenine to the medium (*Figure 5D*). A previous study demonstrated that the addition of 300 μM adenine to the medium increased ATP levels from 4 to 5.5 mM in wild-type cells and from 1 to 4 mM in *bas1Δ* cells (*Gauthier et al., 2008*) (similar results are shown in *Figure 1E*), but induced negligible or no changes in *adk1Δ* cells (from 2 to 2 mM) (*Gauthier et al., 2008*). Consistent with our hypothesis, the addition of adenine-reduced Syn-GFP toxicity and aggregation in *bas1Δ*, but not *adk1Δ* cells (*Figure 5D*). Thus, a high level of ATP prevented Syn-GFP aggregation and toxicity.

We examined another model protein involved in neurodegenerative diseases. PolyQ containing the huntingtin protein is susceptible to aggregation and has been implicated in Huntington's disease (*Poirier et al., 2005*). We investigated the toxicity of Htt103Q, a mutant form of the huntingtin protein that is also susceptible to aggregation and causes cytotoxicity in yeast (*Meriin et al., 2002*). Consistent with the concept that a high level of ATP prevents protein aggregation, the ATP homeostasis mutants *snf1Δ*, *adk1Δ*, *snf1Δ adk1Δ*, and *bas1Δ* were very sensitive to Htt103Q expression (*Figure 5—figure supplement 1*).

## Proteasomes are essential for the removal of protein aggregates induced by ATP depletion

Protein aggregation caused by ATP depletion was cytotoxic (*Figures 3B and 5A*) and was not easily dissolved after ATP repletion (*Figure 4A,B*). To identify a pathway that is essential for the removal of aggregates, we examined the involvement of proteasomes and autophagy.

The deletion of *RPN4*, which encodes a transcription factor of proteasomal genes (*Xie and Varshavsky, 2001*), downregulated proteasomal activity (*Kruegel et al., 2011*) and resulted in synthetic growth defects with *adk1Δ*, *snf1Δ*, *bas1Δ* at a high temperature of 38°C and in the presence of $H_2O_2$ (*Figure 5E*). In contrast to proteasomes, autophagy did not appear to have genetic interactions with the above mutants. The deletion of an essential component of the autophagic pathway, *ATG1* did not affect the sensitivity of *adk1Δ*, *snf1Δ*, *bas1Δ* to a high temperature of 38°C or to $H_2O_2$ (*Figure 5E*). We also did not observe the accumulation of Hsp104-GFP foci in the autophagy mutants *atg1Δ*, *atg8Δ*, and *atg13Δ* (not shown).

To investigate the involvement of proteasomes in the removal of protein aggregates after the transient depletion of ATP, we pretreated cells with the proteasomal inhibitor MG132 or dimethylsulfoxide

(DMSO) and examined the kinetics for the formation of Hsp104-GFP foci after the 2DG treatment (*Figure 5F*) using the drug-sensitive yeast strain Y13206 (*Piotrowski et al., 2017*). Under both conditions, more than 90% of cells exhibited Hsp104-GFP foci within 30 min of the 2DG treatment. More than two-thirds of Hsp104-GFP foci dissolved in the DMSO control, while less than one-third dissolved in MG132-treated samples, indicating that proteasomes are required for the dissolution process (*Figure 5F*).

## Simultaneous imaging of ATP levels and protein aggregates reveals the spontaneous and slow accumulation of protein aggregates in cells with stable ATP Dynamics

To examine the relationship between ATP fluctuations and protein aggregation more directly, we performed simultaneous observations of ATP levels and protein aggregates in wild-type and *snf1Δ adk1Δ* cells. Since the QUEEN construct is optimized for use at 25°C, we took advantage of Hsp104 tagged with RedStar2, a red fluorescent protein optimized for use in budding yeast (*Janke et al., 2004*) (Hsp104-RS2), which visualizes protein aggregation at approximately 25°C. The percentage of cells showing Hsp104-RS2 foci at 25°C and the mean fluorescence intensity of these foci were significantly higher in *snf1Δ adk1Δ* mutant cells than in wild-type cells (*Figure 6A, B* and *Figure 6—figure supplement 1A*). These results suggest that protein aggregates are more likely to accumulate in double mutant cells at 25°C, similar to 37°C (*Figure 3C, D*).

We then conducted the simultaneous time-lapse imaging of QUEEN and Hsp104-RS2 in wild-type cells. *Figure 6C, D* shows an example of the results obtained (four other examples of numerical data are shown in *Figure 6—figure supplement 1B*). All wild-type cells (20 out of 20 cells) showed stable ATP dynamics with a small fluctuation (as shown in *Figure 2A, F*) as well as a slow and linear increase in the intracellular fluorescence intensity of Hsp104-RS2 (which correlated with the number and size of foci) with time. The mean ± standard deviation (SD) of the linear correlation coefficient was 0.99 ± 0.01 (*N* = 20), suggesting that the linear increase in Hsp104 aggregates was robust in cells with stable ATP levels.

As shown in *Figure 2F*, 65% of *snf1Δ adk1Δ* cells did not exhibit an obvious ATP dip or shift. We also performed the simultaneous time-lapse imaging of QUEEN and Hsp104-RS2 in *snf1Δ adk1Δ* cells showing stable ATP dynamics (*Figure 6E, F*, four other examples of numerical data are shown in *Figure 6—figure supplement 2A*). Similar to wild-type cells, all mutant cells (38 out of 38 cells) showed a slow and linear increase in the intracellular fluorescence intensity of Hsp104-RS2 with time. The mean ± SD of the linear correlation coefficient was 0.97 ± 0.02 (*N* = 38). These results suggest that the slow and linear increase in Hsp104 aggregates is a common feature of cells with stable ATP dynamics.

We quantified fluctuations in the mean fluorescence intensity of Hsp104-RS2 by calculating the root mean square deviation (RMSD). A detailed method for this calculation is described in Methods. In brief, differences in fluorescence intensity between the measured values and the linear regression were squared, averaged over a period of time, and the positive square root was taken. We calculated the RMSD in wild-type cells and *snf1Δ adk1Δ* cells over the entire observation period (the mean ± SD was 350 ± 19 min for wild-type cells and 571 ± 70 min for *snf1Δ adk1Δ* mutant cells). The RMSD was significantly larger in mutant cells than in wild-type cells (*Figure 6—figure supplement 2B*). The larger RMSD in mutant cells may reflect the linear increase in the intensity of Hsp104-RS being slightly perturbated by larger ATP fluctuations than in wild-type cells; however, these cells did not exhibit an obvious ATP dip during the observation period.

## The transient decrease in ATP levels is closely associated with the increased accumulation of protein aggregates in *snf1Δ adk1Δ* cells

We further performed the simultaneous time-lapse imaging of QUEEN and Hsp104-RS2 in *snf1Δ adk1Δ* cells that showed an ATP dip (*Figure 7A, B* shows an example, five other examples of numerical data are shown in *Figure 7—figure supplement 1A–C*). A sharp elevation in the intracellular fluorescence intensity of Hsp104-RS2 was noted from a slow and linear increase within hours of the ATP dip. We also quantitatively and statistically examined the sharp elevation in the fluorescence intensity of Hsp104-RS2 followed by the ATP dip by calculating the RMSD. In these cases, the RMSD was calculated using the linear regression of Hsp104-RS2 fluorescence intensities before the onset

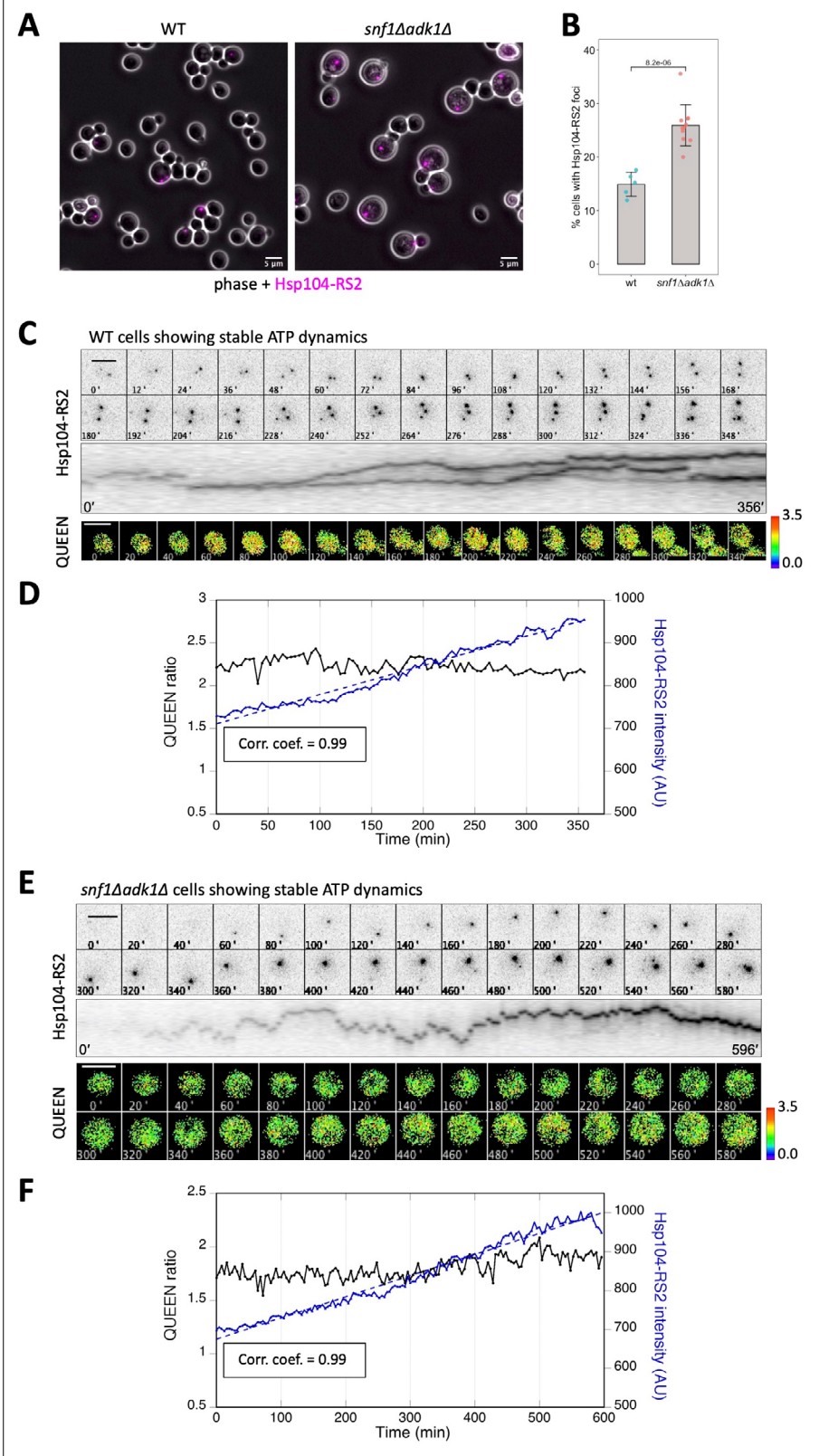

**Figure 6.** The intracellular fluorescence intensity of Hsp104-RS2 increased linearly and slowly in cells showing stable adenosine triphosphate (ATP) dynamics. (**A**) Formation of Hsp104-RS2 foci in wild-type and *snf1Δ adk1Δ* cells. The RFP signal (magenta) was imaged in log-phase cells grown at 25°C in medium containing 2% glucose. (**B**) Fraction of cells showing Hsp104-RS2 foci. The percentages of cells with Hsp104-RS2 foci per one field of view

*Figure 6 continued on next page*

*Figure 6 continued*

(containing 70–342 cells) were plotted. Bars and error bars indicate the mean ± 1SD. The significance of differences was tested using the unpaired two-tailed Welch's *t*-test and indicated by the p value. (**C**) Time-lapse imaging of QUEEN and Hsp104-RS2 in a wild-type cell. (*Top*) Images of the Hsp104-RS2 signal (inverted grayscale) in the cell at the indicated time points are shown. (*Middle*) Kymograph of the images shown in the top panel. (*Bottom*) QUEEN ratio images of the cell. (**D**) The mean QUEEN ratio and the mean fluorescence intensities of Hsp104-RS2 inside the cell were plotted over time. The dotted line indicates the best linear regression of Hsp104-RS2 intensities with a correlation coefficient of 0.99. (**E**) Time-lapse imaging of QUEEN and Hsp104-RS2 in an *snf1Δ adk1Δ* cell showing stable ATP dynamics. (*Top*) Images of the Hsp104-RS2 signal (inverted grayscale) in the cell at the indicated time points are shown. (*Middle*) Kymograph of the images shown in the top panel. (*Bottom*) QUEEN ratio images. (**F**) The mean QUEEN ratio and the mean fluorescence intensities of Hsp104-RS2 inside the cell were plotted over time. The dotted line indicates the best linear regression of Hsp104-RS2 intensities with a correlation coefficient of 0.99. Scale bar = 5 μm.

The online version of this article includes the following source data and figure supplement(s) for figure 6:

**Source data 1.** Raw data for *Figure 6*.

**Figure supplement 1.** Simultaneous imaging of adenosine triphosphate (ATP) and protein aggregation in wild-type cells.

**Figure supplement 1—source data 1.** Raw data for *Figure 6—figure supplement 1*.

**Figure supplement 2.** Simultaneous imaging of adenosine triphosphate (ATP) and protein aggregation in *snf1Δ adk1Δ* cells showing stable ATP levels.

**Figure supplement 2—source data 1.** Raw data for *Figure 6—figure supplement 2*.

of the ATP dip (the dotted line in *Figure 7B*). The periods in which the RMSD before the ATP dip was calculated were more than 92 min, and 248 min on average, ensuring that the fluorescence intensity of Hsp104-RS2 had stably and linearly increased in the hours preceding the ATP dip in these cells. *Figure 7C* shows a line chart of the RMSD before and after the ATP dip (*N* = 25 cells). The RMSD increased by 5.6-fold on average after the dip, and 60% of cells showed more than a 3-fold increase (corresponding to more than three sigma) (*Figure 7D*). The increase observed in the RMSD was already significant 30 min after the ATP dip. Moreover, no significant difference was noted in the RMSD between cells showing stable ATP and those before the ATP dip (*Figure 7—figure supplement 2*). This result suggests that even in double mutant cells, which have the potential to undergo the ATP dip, the intensity of Hsp104-RS2 increased slowly and linearly, similar to that in cells with stable ATP dynamics, until they exhibited the ATP dip. Therefore, these results indicate that the sharp elevation observed in the fluorescence intensity of Hsp104-RS2 is a phenomenon specific to cells exhibiting the ATP dip. Collectively, these results revealed that the ATP dip is closely associated with and often precedes the accelerated accumulation of protein aggregates, suggesting the possibility that the former induces the latter.

## Discussion

In the present study, we demonstrated for the first time that the Snf1 complex, budding yeast AMPK, is required for the stable maintenance of cellular ATP levels (ATP homeostasis) in collaboration with Adk1 (*Figure 8A*). This function of the Snf1 complex in ATP homeostasis is independent of glucose levels in the medium and Mig1, the major transcriptional repressor involved in glucose repression (*Figure 1—figure supplement 1D*); therefore, this is distinct from its well-characterized role in adaptation to glucose limitations. The activity of the Snf1 kinase complex may be sharply tuned depending on the intracellular levels of adenine nucleotides or other metabolites indicative of cellular energy to prevent a rapid ATP dip (*Figure 2*), even in the presence of sufficient amounts of glucose. It is important to note that the reductions observed in intracellular ATP levels in *snf1Δ* cells in the presence of glucose were overlooked in previous biochemical analyses, again demonstrating the usefulness of QUEEN-based ATP imaging. A recent study showed that the deletion of *SNF1* decreased glycolytic flux in cells cultured in 1% glucose (*Martinez-Ortiz et al., 2019*), which supports the novel function of the Snf1 complex.

Since the deletion of *BAS1* induced the greatest reduction in ATP levels and is epistatic to *snf1Δ*, a large pool size of adenine nucleotides is a prerequisite for ATP homeostasis. This assumption is

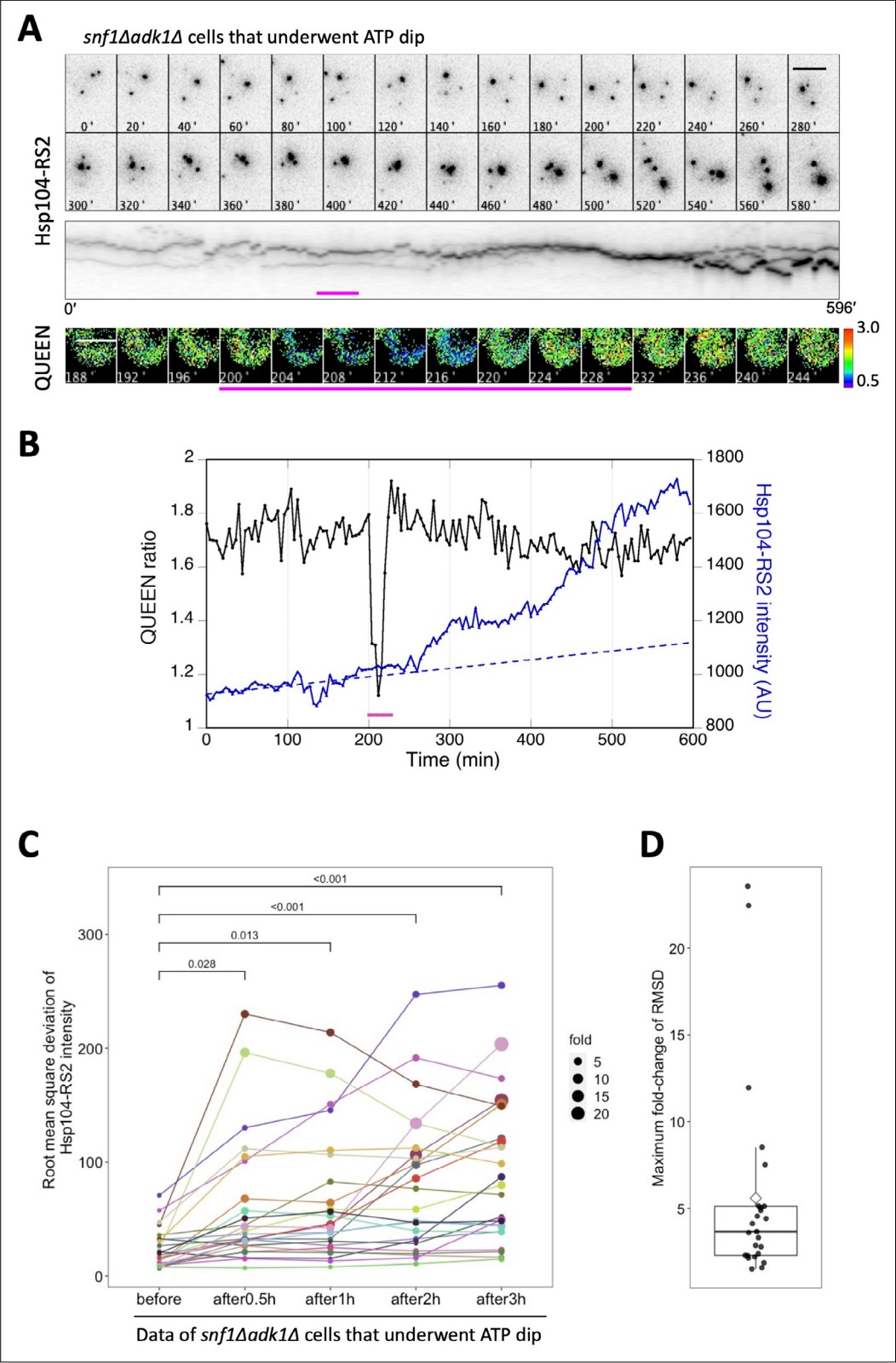

**Figure 7.** The transient decrease in adenosine triphosphate (ATP) levels is closely associated with the increased accumulation of protein aggregates in *snf1Δ adk1Δ* cells. (**A**) Time-lapse imaging of QUEEN and Hsp104-RS2 in an *snf1Δ adk1Δ* cell that underwent a single ATP dip. (*Top*) Images of the Hsp104-RS2 signal (inverted grayscale) in the cell at the indicated time points are shown. (*Middle*) Kymograph of the images shown in the top

*Figure 7 continued on next page*

*Figure 7 continued*

panel. (*Bottom*) QUEEN ratio images of the cell during the ATP dip. (**B**) The mean QUEEN ratio and the mean fluorescence intensities of Hsp104-RS2 inside the cell were plotted over time. The dotted line indicates the best linear regression of Hsp104-RS2 intensities before the onset of the ATP dip ($t$ = 200 min). Magenta bars indicate the duration of the ATP dip. Scale bar = 5 µm. (**C**) The root mean square deviation (RMSD) of the mean Hsp104-RS2 intensity of *snf1Δ adk1Δ* cells that underwent the ATP dip ($N$ = 25 cells, pooled from four independent experiments) were plotted before and after the ATP dip. The RMSD was calculated from deviations from the linear regression before the ATP dip (see Materials and methods for more details). The size of a data point indicates the value normalized to its initial value. The significance of differences was tested using Dunnett's multiple comparison test and indicated by p values. (**D**) Maximum fold changes in RMSD within 3 hr of the ATP dip were box plotted. The mean (a diamond) and median (the median line in the box) were 5.6 and 3.7, respectively.

The online version of this article includes the following source data and figure supplement(s) for figure 7:

**Source data 1.** Raw data for *Figure 7*.

**Figure supplement 1.** Simultaneous imaging of adenosine triphosphate (ATP) and protein aggregation in *snf1Δ adk1Δ* cells that underwent an ATP dip.

**Figure supplement 1—source data 1.** Raw data for *Figure 7—figure supplement 1*.

**Figure supplement 2.** Comparison of the root mean square deviation (RMSD) of the intracellular fluorescence intensity of Hsp104-RS2 in *snf1Δ adk1Δ* cells.

**Figure supplement 2—source data 1.** Raw data for *Figure 7—figure supplement 2*.

reasonable because the pool size of recyclable ATP restricts ATP levels based on the rapid turnover rate of ATP. Bas1 maintains the pool size of ATP by balancing ATP synthesis and irreversible decreases, such as incorporation into RNA and DNA (following conversion to deoxy-ATP), degradation, and excretion in rapidly proliferating yeasts.

The decreases observed in ATP levels in the ATP-mutant cells were confirmed by our biochemical measurements (*Figure 1* and *Figure 1—figure supplement 3*). The biochemical assay also revealed that ADP levels decreased in ATP mutants, similar to ATP levels, and, as a consequence, ATP/ADP ratios, indicators of cellular energy charges, remained largely unchanged. On the other hand, the sum of ATP and ADP levels, indicators of the pool size of adenine nucleotides, decreased in the mutants, which may explain the instability observed in ATP levels. While cytosolic ATP levels were reduced and unstable in *snf1Δ adk1Δ* cells, high ATP levels may have accumulated in intracellular membrane compartments (e.g., vacuoles, lysosomes, and mitochondria), which were not visualized by QUEEN. The hidden fraction of ATP may have increased the average ATP level of whole cells measured by the biochemical assay (*Figure 1C*).

We also showed that key regulators of ATP homeostasis play roles in preventing cytotoxic protein aggregation in budding yeast (*Figure 8B*). The common feature associated with these mutants is reduced ATP levels, suggesting that high ATP levels are essential for protein solubilization.

A proteomic study suggested that the main role of ATP changes depending on its level. At levels lower than 0.5 mM, ATP mainly serves as a substrate for enzymes, such as protein kinases and heat shock protein chaperones, whereas at levels higher than 2 mM, ATP may exert solubilizing effects on disordered proteins (*Sridharan et al., 2019*). ATP homeostasis may also be required to constantly drive proteasomal protein degradation, which requires high levels of ATP (*Benaroudj et al., 2003*; *Tanaka et al., 1983*).

Previous biochemical measurements indicated that although ATP levels were lower in *adk1* and *bas1* mutants than in the wild type, these mutants still had ATP levels that were higher than 2 mM (*Gauthier et al., 2008*). This does not directly explain the accumulation of protein aggregates in these mutants (*Figure 3*) because most proteins are expected to be soluble at >2 mM ATP (*Sridharan et al., 2019*). In this study, we explained this discrepancy by using the biosensor-based ATP imaging technique we developed (*Takaine et al., 2019*; *Yaginuma et al., 2014*). The simultaneous visualization of ATP dynamics and protein aggregates in living cells at the single-cell level revealed that ATP underwent transient depletion (the ATP dip) in AMPK and ADK double mutants. Moreover, the ATP dip strongly correlated with and was often followed by the accelerated accumulation of protein aggregates (*Figure 7*). This result implies that even a transient decrease in cellular ATP levels can trigger the aberrant accumulation of protein aggregates.

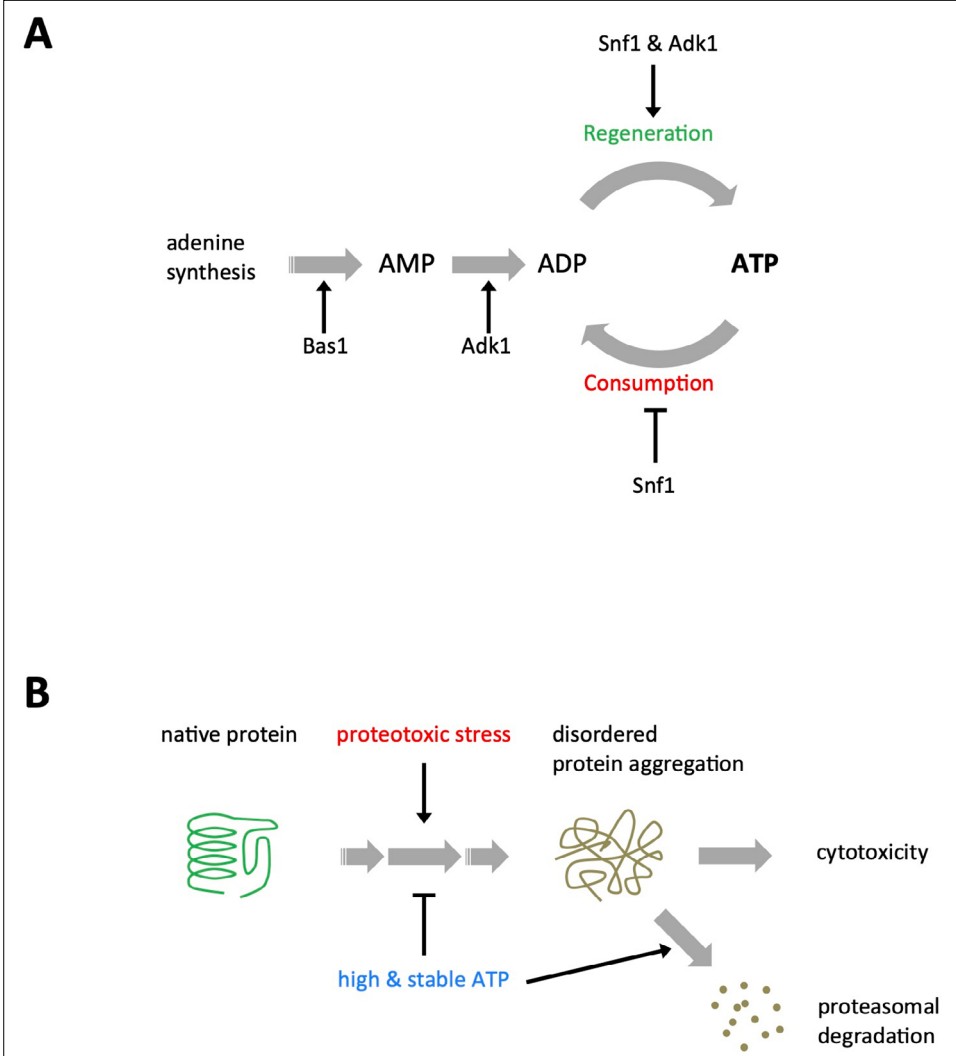

**Figure 8.** Models for adenosine triphosphate (ATP) homeostasis and its role in proteostasis. (**A**) Schematic summary of the roles of Snf1, Adk1, and Bas1 in ATP homeostasis. (**B**) A schematic model for ATP homeostasis preventing cytotoxic protein aggregation.

The ATP dip at 25°C was often followed by the accelerated accumulation of protein aggregates labeled with Hsp104-RS2 within a few hours, but not immediately. We speculate that this was because markedly lower amounts of proteins aggregated upon ATP depletion at 25°C, a temperature at which they were thermodynamically more stable, than at 35°C and the expression level of Hsp104 was also more than 50-fold lower (*Parsell et al., 1994*). Undetectable fine aggregates gradually grow into evident Hsp104-RS2 foci within hours and facilitate the accumulation of whole Hsp104-RS2 foci. Although the mechanisms by which protein aggregates irregularly induced by the ATP dip grow and enhance the overall accumulation of protein aggregates have not yet been elucidated, these results provide insights into how aggregation-prone proteins aggregate and cause cytotoxicity in the mutant. A similar, but distinct, instability in ATP was observed in *bas1Δ*, further confirming that the ATP dip promotes protein aggregation. Although severe ATP depletion in these mutants was gradually recovered by as yet uncharacterized negative feedback regulation, the duration period of ~15 min with reduced ATP levels may induce some proteins to form aggregates that last for generations. The close relationship between uncontrolled fluctuations in ATP levels, rather than a low ATP level itself, and unfavorable protein aggregation again suggests the biological importance of ATP homeostasis. However, the more instantaneous ATP dip that occurred within only seconds or a few minutes, which was unobservable with the current imaging system, may affect protein aggregation in ATP mutants.

A recent study revealed that forced energy (ATP) depletion from budding yeast cells induced extensive cytoplasmic reorganization, including increases in macromolecular (ribosome) crowding, the emergence of numerous membrane-less organelles, and the polymerization of eukaryotic translation initiation factor 2B (*Marini et al., 2020*). Another recent study showed that the depletion of ATP triggered the 'viscoadaptation' of yeast cytoplasm by an as yet unknown mechanism, and induced increases in viscosity and decreases in biomolecular solubility, thereby enhancing phase separation (*Persson et al., 2020*). These findings imply that malfunctional ATP homeostasis may induce undesirable and dysregulated biomolecular assemblies driven by enhanced liquid–liquid phase separation and molecular crowding. The present study using ATP-mutant cells is the first to demonstrate the consequence of failed ATP homeostasis under physiologically relevant conditions and highlights the biological significance of ATP homeostasis. In bacterial cells entering dormancy after the application of antibiotics, the depletion of ATP was shown to induce cytoprotective protein aggregation that regulated the depth of dormancy (*Pu et al., 2019*). Therefore, ATP-dependent protein solubilization/desolubilization may have diverse roles depending on the biological context.

Many neurodegenerative diseases, such as Alzheimer's, Huntington's, and Parkinson's diseases, are associated with protein aggregation (*Eisele et al., 2015*; *Kirkwood et al., 2017*). On the other hand, based on a large body of evidence, mitochondrial dysfunction and accompanying energy failure in nerve cells may result in many types of neurodegenerative diseases (*Haelterman et al., 2014*; *Pathak et al., 2013*). Previous studies demonstrated that ATP levels in the brain were decreased in patients with early Parkinson's disease (*Mochel et al., 2012b*) and also that ATP synthesis in the brain was not properly regulated in patients with early Huntington's disease (*Hattingen et al., 2009*) and in the corresponding mouse model (*Mochel et al., 2012a*). Therefore, protein aggregation induced by the ATP dip, as revealed in the present study, may link energy failure and protein aggregation, providing a comprehensive insight into the onset of neurodegenerative diseases. Further studies are warranted to clarify whether the ATP dip also occurs in the neurons of patients at risk of neurodegenerative diseases or in the elderly. However, neither biochemical measurements nor mass spectrometry is capable of detecting the ATP dip because of their insufficient time and space resolution. Therefore, an ATP imaging approach using the yeast system will be the leading model for elucidating the molecular mechanisms underlying ATP homeostasis and ATP dip-induced protein aggregation.

A recent study reported that the activation of AMPK by metformin ameliorated the progression of Huntington's disease in a mouse model (*Arnoux et al., 2018*), and the potential therapeutic use of metformin for neurodegenerative diseases is being discussed (*Rotermund et al., 2018*). Furthermore, the involvement of ATP and ADK in preventing the manifestation of Parkinson's disease in mouse models and patients has been proposed (*Garcia-Esparcia et al., 2015*; *Nakano et al., 2017*). Protein aggregation induced by the ATP dip may be a general mechanism for the development of proteinopathies. The present study using ATP imaging revealed a physiological consequence of a failure in ATP homeostasis in living cells for the first time and suggests that ATP homeostasis has potential as a target for preventing/treating neurodegenerative diseases.

## Materials and methods
### Yeast strains and plasmids
The budding yeast strains and plasmids used in the present study are listed in *Supplementary files 1 and 2*, respectively. These strains were constructed by a PCR-based method (*Janke et al., 2004*) and genetic crosses. The yeast knockout strain collection was originally purchased from GE Healthcare (cat. # YSC1053). Strains and plasmids used in the present study will be available from the Yeast Genetic Resource Centre Japan (YGRC, http://yeast.nig.ac.jp/yeast/top.xhtml).

### Media and cell culture
The standard technique for the yeast culture and manipulation was used (*Guthrie and Fink, 2002*). Synthetic medium (SC) was prepared as described by *Hanscho et al., 2012*. 2-Deoxy-D-glucose (2DG), tunicamycin, and MG132 were purchased from FUJIFILM Wako (cat. # 046-06483, 202-08241, and 139-18451, respectively). Tunicamycin and MG132 were dissolved in DMSO to make stock solutions (5 mg/ml and 42 mM, respectively). Cells were grown to the mid-log phase at 30°C in SC before imaging unless otherwise noted.

## Microscopy

Cells expressing Hsp104-GFP, Hsp104-RedStar2 (Hsp104-RS2), or GFP-Ubc9ts were concentrated by centrifugation and sandwiched between a slide and coverslip (No. 1.5 thickness, Matsunami, Osaka, Japan). Immobilized cells were imaged using an inverted fluorescent microscope (Eclipse Ti-E, Nikon) equipped with an Apo TIRF ×100 Oil DIC N2/NA 1.49 objective lens and electron-multiplying charge-coupled device camera (iXon3 DU897E-CS0-#BV80, Andor) at approximately 25°C. The Hsp104-GFP, Hsp104-RS2, and GFP-Ubc9ts fluorescent signal was collected from stacks of 11 *z*-sections spaced by 0.5 µm, and the maximum projections of the optical sections were shown in *Figures 4–7* and *Figure 4—figure supplement 1*. Cells expressing QUEEN were immobilized on a concanavalin A-coated 35 mm glass-bottomed dish (#3971-035, No. 1.5 thickness, IWAKI). The dish was filled with an excess amount of medium (4.5–5 ml) against the cell volume to minimize changes in the chemical compositions of the medium during observations. QUEEN fluorescence was acquired as previously described (*Takaine et al., 2019*). Cells expressing Syn-GFP were immobilized on a slide glass as described above, and the fluorescence signal was collected from a single *z*-plane using an inverted fluorescent microscope (Eclipse Ti2-E, Nikon, Tokyo, Japan) equipped with a CFI Plan Apo $\lambda$ ×100 Oil DIC/NA1.45 objective lens and CMOS image sensor (DS-Qi2, Nikon). Images of cells were acquired from several fields of view for each experimental condition. Hsp104-RS2 foci were detected using the FindMaxima function of Fiji software (*Schindelin et al., 2012*) with a noise tolerance set to 4000. The fluorescence intensities of Hsp104-RS2 foci were assessed by measuring mean intensity within a circle with a diameter of 3 pixels centered on the focus.

## Biochemical measurements of ATP and ADP

Whole cell extracts were prepared according to *Seo et al., 2017* with slight modifications. Mid-log cells were harvested in a 1.6-ml microtube by centrifugation and resuspended in 1-ml fresh SC or 40 mM 2DG medium (for ATP depletion). After a 10 min incubation at 30°C, a small fraction of cells (50 µl) was removed for the measurement of cell numbers and optical density, and the remaining cells were pelleted and resuspended in 0.75 ml of 90% acetone. The suspension was incubated at 90°C for 15 min to evaporate acetone. The remaining solution (30–35 µl) was centrifuged at 20,000 × *g* at 4°C for 3 min. The supernatant was mixed with 450 µl of TE (10 mM Tris–HCl, pH 8.0, and 1 mM ethylene-diaminetetraacetic acid). These extracts were stored at −80°C until analyzed. ATP and ADP levels were measured using the ATP determination kit (Invitrogen) and EnzyLight ADP assay kit (EADP-100, Funakoshi), respectively, according to the manufacturers' instructions. Luminescence was measured using an Enspire multimode plate reader (PerkinElmer). All samples were assayed in duplicate. ATP and ADP levels were normalized for an optical density at 600 nm of the initial cell suspension assessed by BioSpectrometer (Eppendorf).

## Data analysis

Numerical data were plotted using KaleidaGraph software ver. 4.5.1 (Synergy Software) and R studio software ver. 3.4.1 (*R Development Core Team, 2017*). Means, SDs, and p values were calculated using Excel software (Microsoft, WA, USA), the KaledaGraph and the R studio. Significance between two sets of data was tested using the unpaired one-tailed Welch's *t*-test unless otherwise noted, and was indicated by an asterisk or p value. For comparison between several mutants and the wild type, statistical significance was tested using Dunnett's multiple comparison test. Data were sometimes represented by a dot plot that shows distribution characteristics in extensive detail. The horizontal bar in the dot plot indicates the average of each population. Box plots show the 75th and 25th percentiles of the data (interquartile range) as the upper and lower edges of the box, the median as the medial line in the box, the 1.5 × interquartile range as whiskers. All measurements were repeated at least twice to confirm reproducibility.

During time-lapse imaging, QUEEN fluorescence was sometimes not acquired by the correct exposure time because of a mechanical malfunction in the microscope system, producing an abnormally high or low QUEEN ratio. The incorrect acquisition of the QUEEN signal was readily noticeable and clearly distinguishable from the ATP dip because one of the two QUEEN fluorescence signals (excited at 410 or 490 nm) was unusually intense in that frame. The error rate was approximately once in 70 frames (which varied day to day), and the abnormal QUEEN ratio was corrected by replacing it with the average value calculated using the previous and next frames.

ATP levels in yeast cells were estimated using QUEEN-based ratiometric imaging, as previously described (*Takaine, 2019*; *Takaine et al., 2019*). The QUEEN ratio is proportional to ATP levels and pseudocolored to reflect its value throughout the present study. The mean QUEEN ratio inside of a cell represents the intracellular ATP level of the cell.

We examined temporal fluctuations in the mean QUEEN ratio inside a cell using time-course data as follows. We calculated differences in the ratio between one frame and the previous frame using the *diff* function of R software ($\Delta ratio\,(t) = ratio_t - ratio_{t-1}$ ($t$ = 1, 2, …, $T$), $T$: the last frame), and found the maximum of differences ($\max\{\Delta ratio(t)\}$). The calculation was then expanded by introducing the parameter $l$ indicating the time lag of the frame to deduct ($\Delta ratio(t, l) = ratio_t - ratio_{t-l}$), and the calculation was repeated. We found that most of the maxima of differences fell between 0.2 and 0.4 in wild-type cells, and the value of the parameter $l$ did not affect the result obtained (*Figure 2—figure supplement 2A*). On the other hand, in *snf1Δ adk1Δ* cells, the maxima of differences were distributed above 0.4 (*Figure 2—figure supplement 2B*), demonstrating large temporal fluctuations in the mean QUEEN ratio specific to the double mutant. In consideration of these results, we defined a transient decrease in the mean QUEEN ratio in a cell of more than 0.5 and spanning more than 2 frames as 'the ATP dip'. If the decrease in the ratio was in the range of 0.33–0.5, it was classified as a shallow ATP dip. In cases in which the mean QUEEN ratio decreased by more than 0.33 and did not recover during the observation period (as shown in *Figure 2—figure supplement 1*), the irreversible change was defined as 'the ATP shift'. If the cell showed neither the ATP dip nor ATP shift during the observation period, at least for the 2 × doubling time (3.6 hr for the wild type and 4.8 hr for *snf1Δ adk1Δ*), the ATP dynamics of the cell were classified as 'stable'.

The RMSD in the mean Hsp104-RS2 intensity of an *snf1Δ adk1Δ* cell undergoing the ATP dip (*Figure 7C* and *Figure 7—figure supplement 2*) was calculated as follows. The time series of the mean Hsp104-RS2 intensity before the ATP dip were fit to a linear regression line, and a deviation was defined as the difference between the measured value of Hsp104-RS2 intensity and the estimated value calculated from the regression line. The time from the start of the time series to the onset of the ATP dip was more than 92 min, and 248 min on average. Deviations were squared and then averaged over the indicated periods: *before*, from the start of the time series to the onset of the ATP dip; *after x h*, $x$ hours after the onset of the ATP dip. RMSD was defined as the positive square root of the average. In calculations of the RMSD of cells that showed stable ATP dynamics without an ATP dip (*Figure 6—figure supplement 2B*), the RMSD was calculated over the entire observation period.

The autocorrelation functions of oscillations in the QUEEN ratio were calculated using R studio software. The apparent period of oscillation was estimated from the positive second peak of the correlation coefficient, which was outside the 95% confidence interval and significant ($p < 0.05$), rejecting the assumption that there is no correlation.

## Acknowledgements

We are grateful to the Yeast Genetic Resource Center, Y Ohya, K Ohashi, H Takagi, D Watanabe, and J Frydman for providing the yeast strains and plasmids. We thank the members of the Yoshida/Takaine laboratories for their support. This work was supported by JSPS grants 16H04781 (SY and MT), 15K18525 (MT), and 19K06654 (MT) and the Takeda Science Foundation (SY). This work was also supported by the joint research program of the Institute for Molecular and Cellular Regulation, Gunma University, Japan.

## Additional information

### Funding

| Funder | Grant reference number | Author |
| --- | --- | --- |
| Japan Society for the Promotion of Science | 19K06654 | Masak Takaine |
| Japan Society for the Promotion of Science | 16H04781 | Masak Takaine Satoshi Yoshida |

| Funder | Grant reference number | Author |
|---|---|---|
| Japan Society for the Promotion of Science | 15K18525 | Masak Takaine |
| Takeda Science Foundation | | Satoshi Yoshida |

The funders had no role in study design, data collection, and interpretation, or the decision to submit the work for publication.

## Author contributions
Masak Takaine, Conceptualization, Data curation, Funding acquisition, Investigation, Methodology, Project administration, Resources, Supervision, Visualization, Writing - original draft, Writing - review and editing; Hiromi Imamura, Conceptualization, Resources, Supervision, Writing - review and editing; Satoshi Yoshida, Conceptualization, Funding acquisition, Project administration, Supervision, Writing - original draft, Writing - review and editing

## Author ORCIDs
Masak Takaine http://orcid.org/0000-0002-1279-9505
Hiromi Imamura http://orcid.org/0000-0002-1896-0443

## Decision letter and Author response
Decision letter https://doi.org/10.7554/eLife.67659.sa1
Author response https://doi.org/10.7554/eLife.67659.sa2

# Additional files

## Supplementary files
• Transparent reporting form

• Supplementary file 1. Strains used in the present study.

• Supplementary file 2. Plasmids used in the present study.

## Data availability
All data generated or analyzed during this study are included in the manuscript and supporting files.

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
