## [Editor Report]

Here the authors apply a live cell imaging technique to monitor changes in [ATP]_cyto_ at the single cell level in yeast. This led to the discovery of marked stability of [ATP]_cyto_ (at millimolar concentration) in wildtype yeast and wild fluctuations in [ATP]_cyto_ in yeasts mutant in AMP kinase and adenylate kinase. The latter fluctuations included deep dips in [ATP]_cyto_, which correlated with enhanced accumulation of model abnormal human disease-associated proteins (α-Synuclein, huntingtin etc). The paper is remarkable for suggesting an important link between cellular energetics and protein folding homeostasis, which may be broadly applicable to cells of diverse phyla.

---

## [Decision Letter]

**Decision letter after peer review:**

Thank you for submitting your article "High and stable ATP levels prevent aberrant intracellular protein aggregation" for consideration by *eLife*. Your article has been reviewed by 3 peer reviewers, one of whom is a member of our Board of Reviewing Editors, and the evaluation has been overseen by a Reviewing Editor and David Ron as the Senior Editor. The following individual involved in review of your submission has agreed to reveal their identity: Liam J Holt (Reviewer #3).

Essential revisions:

1) It is important to co-image the ATP sensor and aggregates in the same cell to directly associate the fluctuations in ATP with aggregation response. At this point the ATP fluctuations are documented and the aggregation is documented but it is not clear these are directly associated. These experiments need to be performed in a sufficient number of cells that a statistically robust correlation is shown.

2) There is a need to show conclusions in a greater number of cells, please increase the N in presented data to improve statistical rigor throughout the paper.

3) The idea that ATP homeostasis is relevant was very exciting to the reviewers but we suggest that a situation with a transient inhibitor of the one of the pathways is performed to assess how acute inhibition differs from the null mutants.

4) We also suggest amplifying the message that the results suggest that fluctuations in ATP may be more relevant than an absolute level of ATP in the cell in the discussion. Please also make all other textual and figure edits suggested below to improve clarity of communication.

*Reviewer #1 (Recommendations for the authors):*

1. What is the pH of the cytosol in the AMK/ADK mutants? Is the mechanism of aggregation during "catastrophe" essentially what has been reported previously in yeast?

2. What happens in a more transient loss of enzyme activity rather than a null?

3. How widespread is the aggregation phenomena- do other proteins aggregate or assemble into filaments as has been reported by others under ATP limitations/stationary phase? It would be good to expand the effect to see if this is similar to what has been seen with other modes of ATP depletion.

4. Are these aggregates asymmetrically segregated if cells are persistently depleted of ATP, such as in the null mutants? It seems if there are persistent spikes in ATP, would see a population of mothers enriched in aggregates and this may have long-term consequences on longevity.

5. If make an extract and add exogenous ATP is this sufficient to rescue the aggregates? This might be a more direct way to assess ATP levels, though will still be an issue to disentangle enzymatic vs direct roles.

*Reviewer #2 (Recommendations for the authors):*

1. Acute activation or inhibition of AMPK with drugs would further test if the effects on ATP levels are direct or indirect (e.g., through transcription or some other downstream effect). For example, it would be nice to see changes in ATP levels and protein aggregation immediately following acute inhibition of AMPK.

2. Figure 2 needs wild-type control images. Furthermore, more quantification of the ATP catastrophe phenotype would strengthen the paper.

3. Figure 3B. Please provide a rationale for why H2O2 is used as an insult.

4. Recheck the statistical tests. I'm not sure a One-tailed Welch test is appropriate in most cases. Most of their data should fit a two-tailed distribution. Also, you can't apply Welch to groups (e.g., Figure 1B); when comparing multiple mutants to wild-type, one must perform a One-Way ANOVA followed by a post-hoc test (Dunnett's multiple comparison test is best when you're comparing several mutants to the wild-type; must test for normality first).

5. Figure 3A. The labels in this figure are so small. I can't read anything.

*Reviewer #3 (Recommendations for the authors):*

1. Based on genetic interaction data, the authors identified protein folding/glycosylation as common genetic interactors with ATP catastrophe mutants. This motivated investigation of several models of protein aggregation (heat shock induced Hsp104 aggregates, neurodegenerative proteins such as α-synuclein and HTT103Q) in ATP mutants. Furthermore, 2DG treatment was also applied to deplete ATP to induce Ubc9ts aggregate formation. However, the authors should be careful when interpreting these data. For example, the authors made use of 2DG in glucose depleted medium to abolish cellular ATP and concluded the Ubc9ts protein aggregates formation were due to ATP removal. However, these treatments can also lead to acidification of cytosol, and whether reduced cytosolic pH contributes to Ubc9ts aggregates formation remains untested. The authors should try buffering the extracellular media to 7.5 to investigate the effect of pH.

2. When the QUEEN data were presented, some graphs were presented as absolute ATP concentrations while others were presented as mean QUEEN ratio. The authors should convert all to absolute ATP concentrations.

3. There are some discrepancies in ATP measurements. For example, comparing the ATP concentrations measured by the QUEEN sensor vs biochemical ascertainment in Figure 1B and 1C, the mutants showed inconsistent levels by the orthogonal methods, yet no explanation or discussion was given.

4. On the other hand, comparing ATP levels in the same condition by QUEEN (WT cells compare Figure 1B and 1E), the WT mean absolute ATP concentration varies around 30%. These two datapoints are essentially biological replicates. This makes me wonder why the authors aren't presenting biological replicates for each experiment.

5. According to early characterization of QUEEN sensor (Yaginuma et al. 2014), the measured 400ex/484ex ratio varies depending on the environmental pH and temperature. Have the authors considered those factors when converting the mean QUEEN ratio to absolute ATP concentration? It would be informative to provide detailed guidelines or conversion tables of the mean QUEEN ratio to absolute ATP concentration. In our opinion, the authors should perform a detailed QUEEN calibration protocol in a supplementary figure and in the Methods. There should be a calibration curve in *S. cerevisiae* cells, rather than reliance on prior results. The autofluorescence properties of *S. cerevisiae* could be quite distinct.

---

## [Author Response]

Essential revisions:1) It is important to co-image the ATP sensor and aggregates in the same cell to directly associate the fluctuations in ATP with aggregation response. At this point the ATP fluctuations are documented and the aggregation is documented but it is not clear these are directly associated. These experiments need to be performed in a sufficient number of cells that a statistically robust correlation is shown.

We appreciate the suggestion. We conducted the simultaneous time-lapse imaging of QUEEN and protein aggregates in wild-type and *snf1∆ adk1∆* cells at 25℃ to examine the relationship between ATP fluctuations and protein aggregation more directly and added the results obtained to Figures 6 and 7 and the accompanying figure supplements. Corresponding information has been added to the figure legends and the Results, Discussion, and Methods sections in the revised manuscript (lines 327-395, lines 451-474, lines 487-490 and lines 623-634). We used Hsp104 tagged with RedStar2 (Hsp104-RS2) to visualize protein aggregates because after several attempts, we found that Hsp104-RS2 labeled protein aggregates at 25℃ (Figure 6A-B), similar to the intense accumulation of aggregates labeled with Hsp104-GFP (Figure 3C), in the double mutant. In cells showing stable ATP levels, either wild-type or *snf1∆ adk1∆*, the intracellular fluorescence intensity of Hsp104-RS2 increased slowly and linearly with time (Figure 6C-F). On the other hand, we found that in *snf1∆ adk1∆* cells that underwent the transient depletion of ATP, namely, the ATP dip (referred as the ATP catastrophe in the previous manuscript), a sharp elevation was observed in the fluorescence intensity of Hsp104-RS2 from the slow and linear increase within hours of the ATP dip (Figure 7). We quantified this sharp elevation in the fluorescence intensity of Hsp104-RS2 by calculating the root mean square deviation (RMSD) (a detailed method for this calculation was described in the Methods section). We then quantitatively and statistically examined the RMSD using data from dozens of examples of wild-type and double mutant cells (Figure 7C and Figure 7—figure supplement 2). These analyses revealed that the sharp elevation in the fluorescence intensity of Hsp104-RS2 was a phenomenon specific to cells exhibiting the ATP dip, and the ATP dip preceded the accelerated accumulation of protein aggregates, suggesting a strong correlation between them. These new results indicate that even a transient decrease in ATP levels can trigger aberrant protein aggregation, again indicating the biological importance of ATP homeostasis.

2) There is a need to show conclusions in a greater number of cells, please increase the N in presented data to improve statistical rigor throughout the paper.

We performed a statistical analysis of ATP dynamics in wild-type and *snf1∆ adk1∆* cells with sufficient sample sizes in revised Figure 2F and Figure 2—figure supplement 2. Corresponding information has been added to the figure legends and the Results and Methods sections in the revised manuscript (lines 176-183 and lines 550-568). In this analysis, we classified ATP dynamics into three types: stable, the ATP dip, and the ATP shift. The ATP dip means a transient decrease in ATP levels, which had been referred to as the ATP catastrophe in the previous manuscript. During the course of the analysis, we found that multiple ATP dips in the same cell were rare and, thus, corrected the text accordingly. The ATP shift means an irreversible decrease in the level of ATP, which we had considered to be nearly equal to cell death in the previous manuscript. In the revised manuscript, we refrained from associating ATP depletion with cell death because based on the findings of experiments conducted in other projects, we found that we cannot confidently assess cell viability according to intracellular ATP concentrations or cell morphologies. Moreover, we reconfirmed statistical rigorousness throughout the revised manuscript (refer to our response to comment #4 by Reviewer #2).

3) The idea that ATP homeostasis is relevant was very exciting to the reviewers but we suggest that a situation with a transient inhibitor of the one of the pathways is performed to assess how acute inhibition differs from the null mutants.

We agree with the comment. However, the acute inhibition of AMPK and ADK at least in budding yeast is practically very difficult for the following reasons:

(i) To the best of our knowledge, there are no inhibitors that specifically block the enzymatic activities of ADK or AMPK of budding yeast.

(ii) Thermo-sensitive mutants of ADK and AMPK have not yet been isolated, and, thus, the acute inactivation of these proteins by a temperature shift is impossible.

(iii) The complete cessation of protein expression generally takes several hours. It seems self-evident that the deletion of ADK1 and BAS1 affects intracellular ATP levels because they have well-defined and limited functions that modulate cellular ATP concentrations; Adk1 catalyzes the interconversion of adenine nucleotides, and Bas1 up-regulates the expression of genes involved in de novo purine synthesis. On the other hand, the Snf1 complex, budding yeast AMPK, clearly appears to be multifunctional, and we reported for the first time a novel function of the Snf1 complex that maintains cellular ATP levels even in the presence of sufficient glucose. However, we are confident of the result because we found that the deletion of each gene encoding the β subunit of AMPK (*SIP1*, *SIP2*, or *GAL83*) reduced cellular ATP levels to a similar extent as the deletion of SNF1. These results support the new function of the SNF1 complex and suggest that three subtypes of the complex are involved in maintaining ATP concentrations. Moreover, a recent study showed that the deletion of SNF1 decreased glycolytic flux in cells cultured in 1% glucose (Martinez-Ortiz et al., 2019, *Yeast*, 36 (8), 487-494), which also supports the function of SNF1. We added these new results to Figure 1—figure supplement 1C and the corresponding information to the Results and Discussion sections (lines 116-119 and lines 359-361).

4) We also suggest amplifying the message that the results suggest that fluctuations in ATP may be more relevant than an absolute level of ATP in the cell in the discussion. Please also make all other textual and figure edits suggested below to improve clarity of communication.

We consider the new results added to Figure 7 to demonstrate the close relationship between the ATP dip and subsequent increase in the accumulation of protein aggregates, and the corresponding discussion (lines 451-474) strengthens our claim that fluctuations in ATP levels may induce cytotoxic protein aggregation. Moreover, we revised the revised manuscript according to the comments suggested below to convey this message more clearly.

Reviewer #1 (Recommendations for the authors):1. What is the pH of the cytosol in the AMK/ADK mutants? Is the mechanism of aggregation during "catastrophe" essentially what has been reported previously in yeast?

According to a genome-wide analysis of the intracellular pH of budding yeast cells, the deletion of *SNF1*, *ADK1*, or *BAS1* does not significantly alter cellular pH (Orij et al., 2012, *Genome Biol.*, 13 (9), R80). Moreover, during the course of revisions, we expressed a protein that assembles into a rod-like structure under low pH conditions in wild-type and *snf1∆ adk1∆* cells and examined its behavior. We found that the marker protein did not show any rod-like structures and diffused in the cytoplasm in both strains (data not shown), also suggesting that intracellular pH in double mutant cells is the same as that in wild-type cells. Based on these results, we consider increased protein aggregation induced by the ATP dip (referred as the ATP catastrophe in the previous manuscript) to be a novel phenomenon reported herein for the first time.

2. What happens in a more transient loss of enzyme activity rather than a null?

Unfortunately, the acute inhibition of AMPK and ADK, at least in budding yeast, is practically very difficult for the following reasons:

(i) To the best of our knowledge, there are no inhibitors that specifically block the enzymatic activities of ADK or AMPK of budding yeast.

(ii) Thermo-sensitive mutants of ADK and AMPK have not yet been isolated, and, thus, the acute inactivation of these proteins by a temperature shift is impossible.

It seems self-evident that the deletion of ADK1 and BAS1 affects intracellular ATP levels because they have well-defined and limited functions that modulate cellular ATP concentrations; Adk1 catalyzes the interconversion of adenine nucleotides, and Bas1 upregulates the expression of genes involved in de novo purine synthesis. On the other hand, the Snf1 complex, budding yeast AMPK, clearly appears to be multifunctional, and we reported for the first time a novel function of the Snf1 complex that maintains cellular ATP levels even in the presence of sufficient glucose. However, we are confident of the result because we found that the deletion of each gene encoding the β subunit of AMPK (*SIP1*, *SIP2*, or *GAL83*) reduced cellular ATP levels to a similar extent as the deletion of SNF1. These results support the new function of the SNF1 complex and suggest that three subtypes of the complex are involved in maintaining ATP concentrations. Moreover, a recent study showed that the deletion of SNF1 decreased glycolytic flux in cells cultured in 1% glucose (Martinez-Ortiz et al., 2019, *Yeast*, **36** (8), 487-494), which also supports the function of SNF1. We added these new results to Figure 1—figure supplement 1C and the corresponding information to the Results and Discussion sections (lines 117-120 and lines 408-410).

3. How widespread is the aggregation phenomena- do other proteins aggregate or assemble into filaments as has been reported by others under ATP limitations/stationary phase? It would be good to expand the effect to see if this is similar to what has been seen with other modes of ATP depletion.

Large-scale studies revealed that many metabolic enzymes in budding yeast assemble into cytoplasmic foci or filaments under nutrient starvation conditions (Narayanaswamy et al., 2009, *PNAS*, 106 (25), 10147-10152; Noree et al., 2010, *JCB*, 190 (4), 541-551; Noree et al., 2019, *MBC*, 30 (21), 2721-2736); however, none have been proven to aggregate through the direct effects of ATP depletion. For example, the glutamine synthetase Gln1 forms filaments upon glucose starvation (Narayanaswamy et al., 2009; Petrovska et al., 2014, *eLife*), but not upon ATP depletion by 2-deoxyglucose (our unpublished observation). On the other hand, Pu et al. (2019) showed that in bacterial cells, a decrease in intracellular ATP levels during nutrient starvation directly promoted the aggregation of endogenous proteins to form “aggresomes” that regulate cell dormancy (Pu et al., 2019, *Molecular Cell*, 73 (1), 143156). The dynamic assembly of bacterial aggresomes is similar to protein aggregation induced by the ATP dip (this study) in that they are both driven by ATP-dependent protein solubilization/desolubilization, which appears to mainly be driven by the hydrotropic effects of ATP (Patel et al., 2017, *Science*, 356 (6339), 753-756). On the other hand, they also have dissimilarities because the assembly of the aggresome is a regulated mechanism and noncytotoxic, whereas the ATP dip irregularly induces cytotoxic protein aggregation. We added a discussion to the revised manuscript (lines 487-490).

4. Are these aggregates asymmetrically segregated if cells are persistently depleted of ATP, such as in the null mutants? It seems if there are persistent spikes in ATP, would see a population of mothers enriched in aggregates and this may have long-term consequences on longevity.

As far as we observed, protein aggregation induced by ATP depletion appeared to be distributed evenly in mother and daughter cells. In addition, we sometimes noted a large Hsp104-RS2 focus that migrated from a mother cell to a daughter cell. These behaviors of protein aggregates appear to differ from those of protein aggregates induced by a high temperature that are retained in mother cells and excluded from daughter cells (Liu et al., 2010, *Cell*, 140 (2), 257-267). The protein aggregates induced by ATP depletion may have biological properties that differ from those of heat-induced aggregates and are separately controlled. In future studies, we will examine the inheritance and clearance of ATP depletion-induced protein aggregation and hope to verify whether aggregates are an aging factor.

5. If make an extract and add exogenous ATP is this sufficient to rescue the aggregates? This might be a more direct way to assess ATP levels, though will still be an issue to disentangle enzymatic vs direct roles.

We may be able to extract intact protein aggregates from yeast cells by preparing a dense native cell lysate that mimics the intracellular milieu (*cf.*, Persson et al., 2020, *Cell*, 183 (6), 1572-1585). As the Reviewer suggested, the addition of exogenous ATP may dissolve aggregates. Possible mechanisms for ATP-induced dissolution include (1) degradation by proteasomes, (2) disaggregation into denatured proteins by Hsp104, and (3) the further refolding of denatured proteins into native proteins by the chaperones Hsp70 and Hsp40. Using specific inhibitors and mutants, we may be able to differentiate these mechanisms and quantitatively examine protein dissolution through each pathway. However, we cannot assess the hydrotropic effects of ATP in the cell-free system because ATP as a hydrotrope may inhibit the denaturation of proteins and subsequent aggregation, but not resolve preexisting protein aggregates (Patel et al., 2017, *Science*, 356 (6339), 753-756). To examine the hydrotropic effects of ATP, ATP concentrations need to be decreased and the de novo assembly of denatured proteins or protein aggregates needs to be detected in the experimental system. These procedures, particularly the rapid depletion of ATP, appear to be difficult in the cell extract system.

Reviewer #2 (Recommendations for the authors):1. Acute activation or inhibition of AMPK with drugs would further test if the effects on ATP levels are direct or indirect (e.g., through transcription or some other downstream effect). For example, it would be nice to see changes in ATP levels and protein aggregation immediately following acute inhibition of AMPK.

We agree with the comment. However, the acute inhibition (or activation) of AMPK and ADK, at least in budding yeast, is practically very difficult for the following reasons:

(i) To the best of our knowledge, there are no inhibitors that specifically block the enzymatic activities of ADK or AMPK of budding yeast.

(ii) Thermo-sensitive mutants of ADK and AMPK have not yet been isolated, and, thus, the acute inactivation of these proteins by a temperature shift is impossible.

(iii) The complete cessation of protein expression generally takes several hours. It seems self-evident that the deletion of ADK1 and BAS1 affects intracellular ATP levels because they have well-defined and limited functions that modulate cellular ATP concentrations; Adk1 catalyzes the interconversion of adenine nucleotides, and Bas1 upregulates the expression of genes involved in de novo purine synthesis. On the other hand, the Snf1 complex, budding yeast AMPK, clearly appears to be multifunctional, and we reported for the first time a novel function of the Snf1 complex that maintains cellular ATP levels even in the presence of sufficient glucose. However, we are confident of the result because we found that the deletion of each gene encoding the β subunit of AMPK (*SIP1*, *SIP2*, or *GAL83*) reduced cellular ATP levels to a similar extent as the deletion of SNF1. These results support the new function of the SNF1 complex and suggest that three subtypes of the complex are involved in maintaining ATP concentrations. Moreover, a recent study showed that the deletion of SNF1 decreased glycolytic flux in cells cultured in 1% glucose (Martinez-Ortiz et al., 2019, *Yeast*, 36 (8), 487-494), which also supports the function of SNF1. We added these new results to Figure 1—figure supplement 1C and the corresponding information to the Results and Discussion sections (lines 117-120 and lines 408-410).

2. Figure 2 needs wild-type control images. Furthermore, more quantification of the ATP catastrophe phenotype would strengthen the paper.

We agree with the comment and added wild-type data to revised Figure 2A. Moreover, we performed a statistical analysis of ATP dynamics in wild-type and *snf1∆ adk1∆* cells with sufficient sample sizes in revised Figure 2F and Figure 2—figure supplement 2. Corresponding information has been added to the figure legends and the Results and Methods sections in the revised manuscript (lines 170-184 and lines 603-621). In this analysis, we classified ATP dynamics into three types: stable, the ATP dip, and the ATP shift. The ATP dip means a transient decrease in ATP levels, which had been referred to as the ATP catastrophe in the previous manuscript. During the course of the analysis, we found that multiple ATP dips in the same cell were rare and, thus, corrected the text accordingly. The ATP shift means an irreversible decrease in the level of ATP, which we had considered to be nearly equal to cell death in the previous manuscript. In the revised manuscript, we refrained from associating ATP depletion with cell death because based on the findings of experiments conducted in other projects, we found that we cannot confidently assess cell viability according to intracellular ATP concentrations or cell morphologies.

3. Figure 3B. Please provide a rationale for why H2O2 is used as an insult.

We added the rationale for using hydrogen peroxide as an inducer of oxidative stress to the revised manuscript (line 222).

4. Recheck the statistical tests. I'm not sure a One-tailed Welch test is appropriate in most cases. Most of their data should fit a two-tailed distribution. Also, you can't apply Welch to groups (e.g., Figure 1B); when comparing multiple mutants to wild-type, one must perform a One-Way ANOVA followed by a post-hoc test (Dunnett's multiple comparison test is best when you're comparing several mutants to the wild-type; must test for normality first).

We considered a one-tailed test to be appropriate in most cases because results are generally predictable based on previous studies (*e.g.*, decrease in ATP level in mutant cells) or direct observations of the data obtained before quantification (*e.g.*, percentages of cells with protein foci). However, we rechecked statistical tests according to the Reviewer’s suggestion and applied two-tailed tests to some cases in the revised manuscript (Figures 5C, 5D, 6B, Figure 1— figure supplement 1C-D, Figure 6—figure supplement 1A and Figure 6—figure supplement 2B) where a two-tailed test was more appropriate. Moreover, we applied Dunnett’s tests for several cases (Figures 1B-E, 3D, 5C, Figure 1—figure supplement 1C-D, Figure 1—figure supplement 3A, C, D, Figure 1—figure supplement 4B, Figure 7C and Figure 7—figure supplement 2) in the revised manuscript according to the Reviewer’s suggestion.

5. Figure 3A. The labels in this figure are so small. I can't read anything.

We labeled the “protein folding” and “protein turn over” categories more clearly in revised Figure 3A. Moreover, in Figure 3—figure supplement 1B in the revised manuscript, we added an enlarged version of the function map of ADK1 shown in Figure 3A and labeled all the categories of functions as a reference for the readers.

Reviewer #3 (Recommendations for the authors):1. Based on genetic interaction data, the authors identified protein folding/glycosylation as common genetic interactors with ATP catastrophe mutants. This motivated investigation of several models of protein aggregation (heat shock induced Hsp104 aggregates, neurodegenerative proteins such as α-synuclein and HTT103Q) in ATP mutants. Furthermore, 2DG treatment was also applied to deplete ATP to induce Ubc9ts aggregate formation. However, the authors should be careful when interpreting these data. For example, the authors made use of 2DG in glucose depleted medium to abolish cellular ATP and concluded the Ubc9ts protein aggregates formation were due to ATP removal. However, these treatments can also lead to acidification of cytosol, and whether reduced cytosolic pH contributes to Ubc9ts aggregates formation remains untested. The authors should try buffering the extracellular media to 7.5 to investigate the effect of pH.

According to a genome-wide analysis of the intracellular pH of budding yeast cells, the deletion of *SNF1*, *ADK1*, or *BAS1* does not significantly alter cellular pH (Orij et al., 2012, *Genome Biol.*, 13 (9), R80). Moreover, during the course of revisions, we expressed a protein that assembles into a rod-like structure under low pH conditions in wild-type and *snf1∆ adk1∆* cells and examined its behavior. We found that the marker protein did not show any rod-like structures and diffused in the cytoplasm in both strains (data not shown), also suggesting that intracellular pH in double mutant cells is the same as that in wild-type cells. Based on these results, we consider increased protein aggregation induced by the ATP dip (referred as the ATP catastrophe in the previous manuscript) to be a novel phenomenon reported herein for the first time.

2. When the QUEEN data were presented, some graphs were presented as absolute ATP concentrations while others were presented as mean QUEEN ratio. The authors should convert all to absolute ATP concentrations.

We represented the data as the mean QUEEN ratio when comparisons of relative values, not absolute ATP concentrations, were sufficient.

3. There are some discrepancies in ATP measurements. For example, comparing the ATP concentrations measured by the QUEEN sensor vs biochemical ascertainment in Figure 1B and 1C, the mutants showed inconsistent levels by the orthogonal methods, yet no explanation or discussion was given.

We discussed the inconsistency of ATP levels in the double mutant in the Discussion section of the revised manuscript (lines 425-429).

4. On the other hand, comparing ATP levels in the same condition by QUEEN (WT cells compare Figure 1B and 1E), the WT mean absolute ATP concentration varies around 30%. These two datapoints are essentially biological replicates. This makes me wonder why the authors aren't presenting biological replicates for each experiment.

Many factors may affect the QUEEN ratio, including the physiological conditions of the cell culture, the batch of the culture medium, temperature, the power output of the fluorescence excitation lamp, and the conditions of optical filters (excitation, emission, and neutral density) for fluorescence microscopy. These factors slightly vary day to day independently of each other and, thus, the combination of their differences may produce a 30% difference in the mean QUEEN ratio even between the same strains. Therefore, we typically compare data obtained from experiments performed on the same day and within the same time zone, and refrain from pooling data from experiments performed on different days (*i.e.*, biological replicates). We confirmed that similar results were obtained from experiments performed on different days.

5. According to early characterization of QUEEN sensor (Yaginuma et al. 2014), the measured 400ex/484ex ratio varies depending on the environmental pH and temperature. Have the authors considered those factors when converting the mean QUEEN ratio to absolute ATP concentration? It would be informative to provide detailed guidelines or conversion tables of the mean QUEEN ratio to absolute ATP concentration. In our opinion, the authors should perform a detailed QUEEN calibration protocol in a supplementary figure and in the Methods. There should be a calibration curve in S. cerevisiae cells, rather than reliance on prior results. The autofluorescence properties of S. cerevisiae could be quite distinct.

As mentioned in the response to comment #1, we assumed that intracellular pH in ATP mutant cells was nearly identical to that in wild-type cells (and bacterial cells). On the other hand, temperature was carefully controlled during the imaging of QUEEN signals because we know that at a temperature higher than 25℃, the QUEEN ratio appears to be low and the dynamic range of the QUEEN ratio narrows. The mean QUEEN ratio was converted to an absolute ATP concentration according to the previously reported procedure (Takaine et al., 2019, *JCS*; Takaine, 2019, *Bio-protocol*). We confirmed that the intensities of autofluorescence in yeast cells were at negligible levels to QUEEN fluorescence. We assumed that the calibration of ATP was appropriate because the estimated values were consistent with intracellular ATP concentrations calculated from biochemical measurements of ATP based on a cell volume of wild-type budding yeast cell of 42 µm^3^ (Jorgensen et al., 2002, *Science*, 297 (5580), 395-400). Detailed guidelines and protocols regarding the QUEEN calibration are needed for future studies that require the more precise estimation of intracellular ATP levels.